# Structural basis for ligand recognition and signaling of hydroxy-carboxylic acid receptor 2

Jae-Hyun Park [1,11], Kouki Kawakami [2,11], Naito Ishimoto [1], Tatsuya Ikuta [2], Mio Ohki[1], Toru Ekimoto[3], Mitsunori Ikeguchi[3,4], Dong-Sun Lee[5], Young-Ho Lee [6,7,8,9,10], Jeremy R. H. Tame [1], Asuka Inoue [2] ✉ & Sam-Yong Park [1] ✉

Hydroxycarboxylic acid receptors (HCAR1, HCAR2, and HCAR3) transduce $G_{i/o}$ signaling upon biding to molecules such as lactic acid, butyric acid and 3-hydroxyoctanoic acid, which are associated with lipolytic and atherogenic activity, and neuroinflammation. Although many reports have elucidated the function of HCAR2 and its potential as a therapeutic target for treating not only dyslipidemia but also neuroimmune disorders such as multiple sclerosis and Parkinson's disease, the structural basis of ligand recognition and ligand-induced $G_i$-coupling remains unclear. Here we report three cryo-EM structures of the human HCAR2–$G_i$ signaling complex, each bound with different ligands: niacin, acipimox or GSK256073. All three agonists are held in a deep pocket lined by residues that are not conserved in HCAR1 and HCAR3. A distinct hairpin loop at the HCAR2 N-terminus and extra-cellular loop 2 (ECL2) completely enclose the ligand. These structures also reveal the agonist-induced conformational changes propagated to the G-protein-coupling interface during activation. Collectively, the structures presented here are expected to help in the design of ligands specific for HCAR2, leading to new drugs for the treatment of various diseases such as dyslipidemia and inflammation.

Humans and other higher primates possess three highly similar G-protein-coupled receptors (GPCRs) in the hydroxy-carboxylic acid (HCA) receptor family, which consists of HCAR1, HCAR2, and HCAR3, also known as GPR81, GPR109A, and GPR109B, respectively[1]. HCAR2

and HCAR3 share 95% sequence identity, differing mainly in the ligand pocket and intracellular C-terminal tail. HCAR2 responds to niacin[2,3] (nicotinic acid, also known as vitamin B3) as well as other ketone body-containing metabolites such as beta-hydroxybutyrate (BHB)[4]. In

[1]Drug Design Laboratory, Graduate School of Medical Life Science, Yokohama City University, Tsurumi, Yokohama 230-0045, Japan. [2]Graduate School of Pharmaceutical Sciences, Tohoku University, Sendai 980-8578, Japan. [3]Computational Life Science Laboratory, Graduate School of Medical Life Science, Yokohama City University, Yokohama City University, Tsurumi, Yokohama 230-0045, Japan. [4]HPC- and AI-driven Drug Development Platform Division, Center for Computational Science, RIKEN, Yokohama 230-0045, Japan. [5]Bio-Health Materials Core-Facility Center and Interdisciplinary Graduate Program in Advanced Convergence Technology and Science, Jeju National University, Jeju 63243, Republic of Korea. [6]Research Center for Bioconvergence Analysis, Korea Basic Science Institute, Ochang, Chungbuk 28119, Republic of Korea. [7]Bio-Analytical Science, University of Science and Technology, Daejeon 34113, Republic of Korea. [8]Graduate School of Analytical Science and Technology, Chungnam National University, Daejeon 34134, Republic of Korea. [9]Department of Systems Biotechnology, Chung-Ang University, Gyeonggi 17546, Republic of Korea. [10]Frontier Research Institute for Interdisciplinary Sciences, Tohoku University, Miyagi 980-8578, Japan. [11]These authors contributed equally: Jae-Hyun Park, Kouki Kawakami. ✉e-mail: iaska@tohoku.ac.jp; park@yokohama-cu.ac.jp

contrast, HCAR1 and HCAR3 respond to different metabolites such as lactic acid or 3-hydroxyoctanoic acid, respectively[5,6]. Niacin has been used for decades as an antiatherogenic drug that is also able to decrease plasma levels of low-density lipoprotein (LDL). Its ability to reduce the production of certain proinflammatory cytokines and macrophage chemotaxis is mediated by HCAR2[7]. Since HCAR2 was believed for many years to help lower the plasma levels of LDL, drug development campaigns focused on the receptor to develop new treatments for dyslipidemia[8,9]. More recently, HCAR2 was shown to mediate neuroprotective effects and inflammatory response regulation, as well as playing a role in the suppression of colorectal cancer[10–13]. HCAR2-knockout mice show a greater susceptibility to colonic inflammation and colon cancer[12]. Although conceived initially as a drug target for the control of lipid metabolism, HCAR2 therefore has potential as a route to novel treatments of a wide range of inflammatory diseases, as well as atherosclerosis, multiple sclerosis and Parkinson's disease, and as a result it is of growing interest for pharmaceutical applications.

Despite the usefulness of niacin in the treatment of atherosclerosis, patient compliance is greatly weakened by cutaneous flushing, caused by HCAR2 triggering the release of vasodilatory prostaglandins from Langerhans cells and keratinocytes[14,15]. Several synthetic agonists of HCAR2 have been developed with a view to

lowering plasma LDL concentrations, including the niacin derivatives acipimox[16], acifran[17] and MK-0354[18], as well as novel compounds such as GSK256073[8] and MK-1903[19]. In 2003, acipimox was approved by the Food and Drug Administration for the treatment of hyperlipoproteinemia. Although acipimox showed higher efficacy against dyslipidemia than niacin, it still caused the side-effect of flushing. Potent HCAR2 agonists were developed after the commercialization of acipimox, but only acipimox and acifran proved able to deliver the desired lipid profile. Although many studies have indicated that HCAR2 is involved in various inflammatory disorders and dyslipidemia, the signaling mechanism of HCAR2 is still unknown at the molecular level. To assist in the understanding of HCAR2 activation and the development of novel HCAR2 agonists, we have determined the structure of the niacin-bound signaling complex by single particle cryogenic electron microscopy (cryo-EM).

## Results

### Overall structures of HCAR2-G$_i$ complex

Using NanoBiT-based functional assays, we confirmed that binding of niacin, acipimox and GSK256073 to HCAR2 induces dissociation of the G$_i$ heterotrimer (Gα$_{i1}$, Gβ$_1$ and Gγ$_2$) and recruitment of β-arrestin1 (Fig. 1a)[20]. For structural studies, we used an N-terminally BRIL-fused HCAR2 construct (see Methods). The NanoBiT assays confirmed that

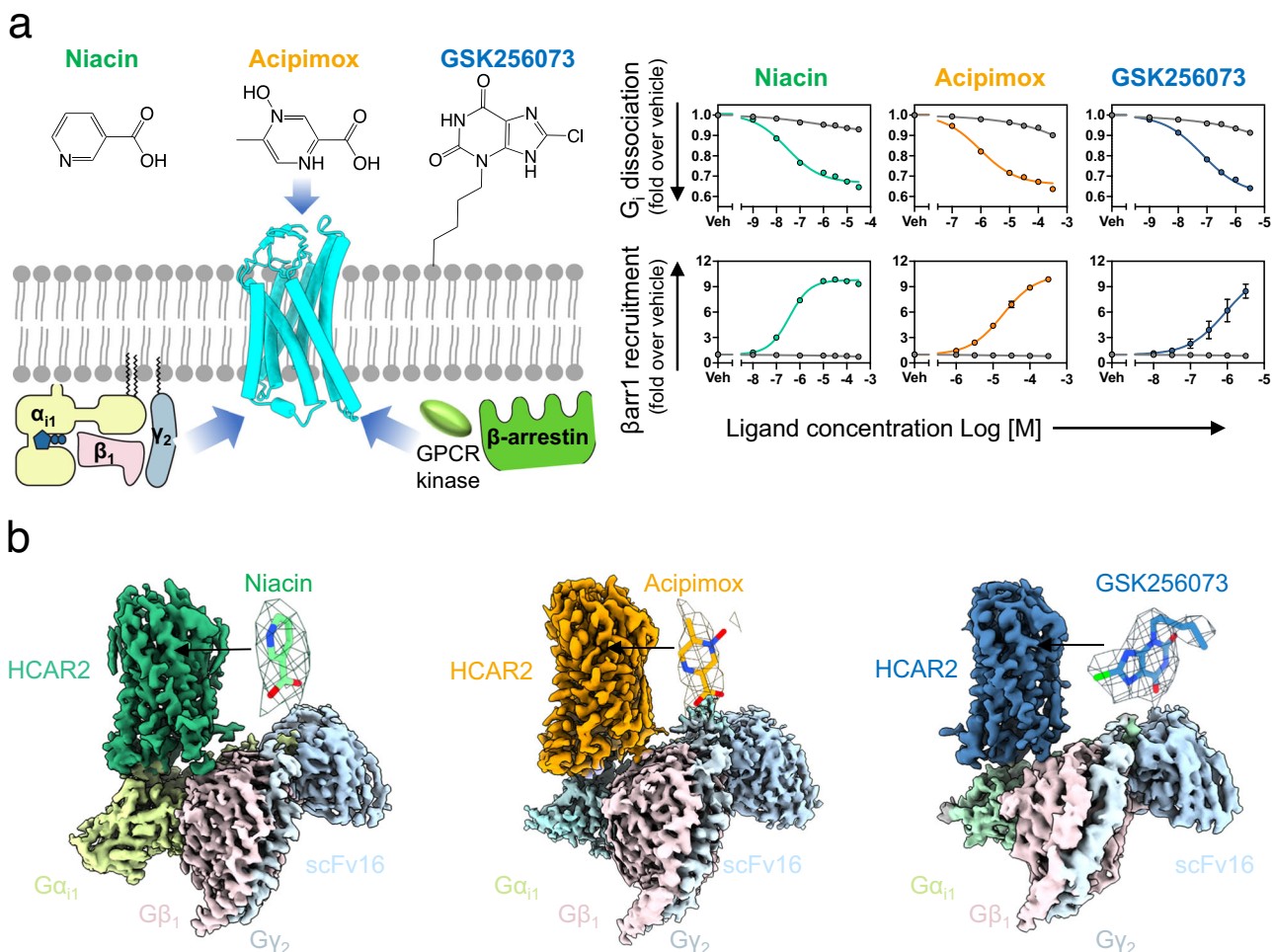

**Fig. 1 | Overall structure and functional analysis of G$_i$-coupled HCAR2 complexes. a** Schematic illustration of agonist binding and signal transductions of HCAR2 (left). The NanoBiT-based assays (right) to measure G$_i$ dissociation and β-arrestin1 (βarr1) recruitment on HCAR2 induced by niacin (green), acipimox (orange) and GSK246073 (blue). Non-specific responses in the mock-transfected cells are shown as gray lines. Circles and error bars represent mean and standard error of the mean (SEM), respectively, of three independent experiments. Error bars are not shown where they are smaller than the circle symbols. **b** Orthogonal views of the density maps of HCAR2-G$_i$ heterotrimer-scFv16 complexes with the ligands niacin (left), acipimox (center) and GSK256073 (right). Maps for receptor and G proteins are from local refinement maps and overall refinement maps, respectively.

the BRIL-HCAR2 construct fully retained the $G_i$ and β-arrestin1 responses to all three ligands when compared with the expression-matched wild-type HCAR2 (Supplementary Fig. 1). This modified human HCAR2 was complexed with $G_i$ heterotrimer stabilized by single chain variable fragment 16 (scFv16) in the presence of agonists (Supplementary Fig. 2). The HCAR2-$G_i$ signaling complexes were subjected to cryo-EM single particle analysis and the models of the complex were refined to a global resolution ranging from 3.1 Å to 3.7 Å (Supplementary Fig. 3 and Supplementary Table 1). The structures displayed clear cryo-EM density for the majority of side-chains and the small-molecule ligand, with the exception of the transmembrane helix 1 (TM1), Gα GTPase domain and N-terminal regions of Gβ and Gγ. (Fig. 1b, Supplementary Fig. 4). As expected of a typical class A GPCR, HCAR2 shows seven transmembrane helices (TM1-7). Additionally, bends are found in TM5, TM6 and TM7 at the conserved proline residues P200[5.50], P246[6.50] and P291[7.50] (Ballesteros-Weinstein numbering is shown as superscripts) (Fig. 2a). Three disulfide bonds are found within the N-terminus and extracellular loops (ECL) (Fig. 2b). Cys18[N-term] and Cys19[N-term] of the N-terminal region form bonds with C183[ECL2] and C266[ECL3], respectively. An additional bond is seen between C100[3.25] (at the N terminus of TM3) and C177[ECL2]. The presence of P168[ECL2] and a beta hairpin loop further reduce the flexibility of ECL2 (Fig. 2b, c). These structural features isolate the ligand binding pocket of HCAR2 from external solvent, unlike most class A GPCRs (Fig. 2c and

Supplementary Fig. 5a). As with other typical class A GPCRs, the membrane-embedded region of HCAR2 is strongly hydrophobic (Fig. 2d). Within TM3, TM4, and TM5, a distinctive pocket is formed by hydrophobic residues, which are highly conserved to those in CX3CR1 forming the binding site of cholesterol; cryo-EM density within this pocket suggests potential binding of cholesterol[21] (Fig. 2e). The models of HCAR2 complexed with niacin, acipimox or GSK256073 are consistent with the overall features observed in many other activated class A GPCRs bound to a G-protein heterotrimer, and almost identical with each other. The niacin complex gives Root Mean Square Deviation (RMSD) values of only 0.59 Å or 0.79 Å for all 291 Cα atoms with the acipimox and GSK256073 complexes respectively. In all three models of HCAR2, the TM helices overlay closely but there are minor differences.

## Orthosteric binding site

The ligand binding pocket lies deeply buried within the HCAR2 structure, surrounded by TM2, TM3 and TM7, with ECL2 acting as a lid for the orthosteric binding site (Fig. 2b–d). W91[ECL1] presses against the ligands, helping to hold them in a stable pose (Figs. 2c, 3a–c). Niacin and acipimox closely resemble each other and make similar interactions with the protein, principally a salt bridge between R111[3.36] and the ligand carboxyl group (Fig. 3a, b). Niacin forms a hydrogen bond with the side-chain of Y87[2.64] through its nitrogen atom within the ring

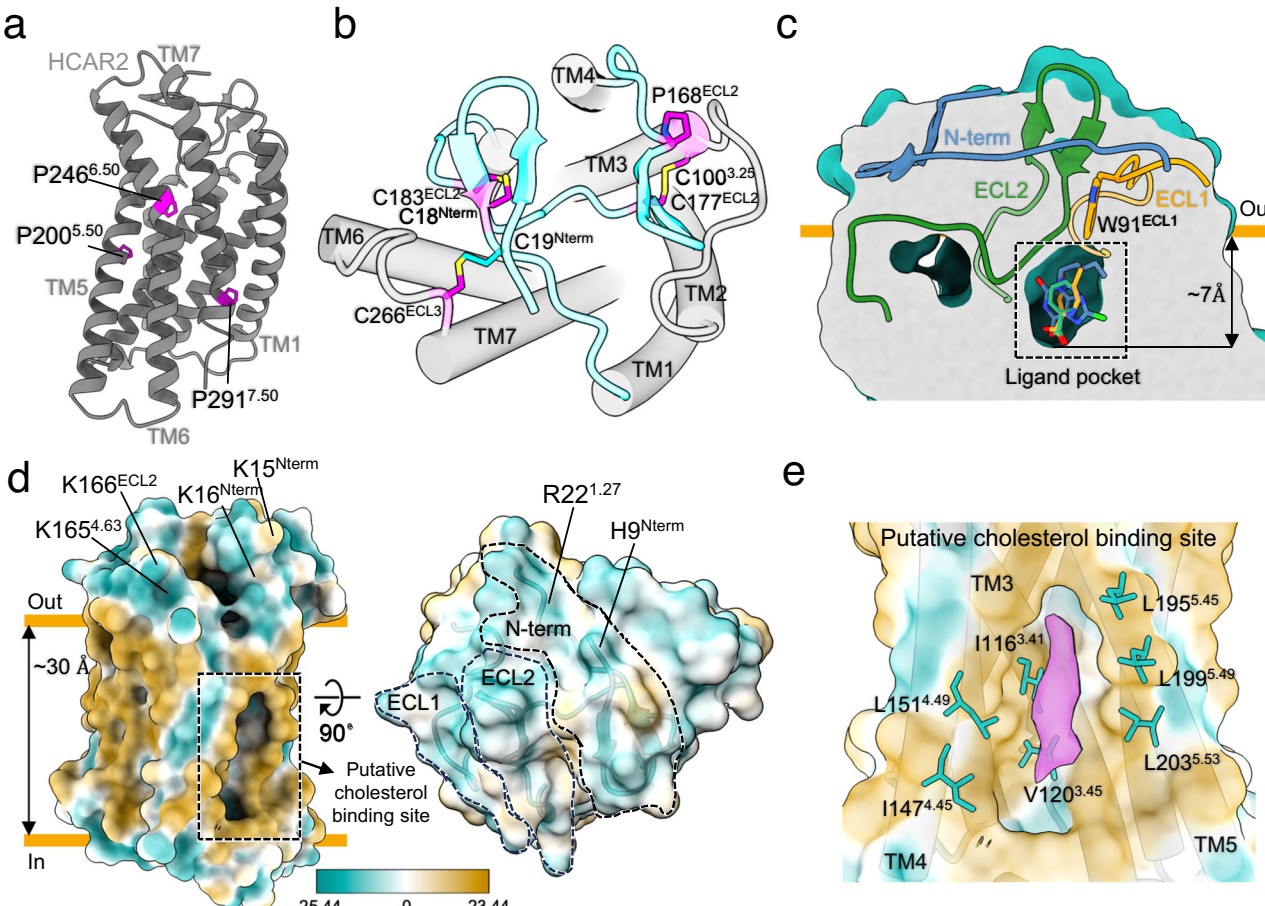

**Fig. 2 | Structural features of active HCAR2. a** The overall structure of HCAR2. Well conserved proline residues are shown as pink stick models and labeled. **b** Lid structure for the orthosteric binding is shown as a cyan cartoon model. Three disulfide bonds and proline residues, which supply rigidity on the extracellular face of HCAR2, are shown as pink cartoon and stick models. Disulfide bonds are colored yellow. **c** Section through the receptor illustrating the orthosteric ligand binding site and the lid structure of N-terminus, ECL1 and ECL2. **d** Molecular surface of HCAR2 (hydrophobic and hydrophilic areas are shown as yellow and blue, respectively). The putative binding site of cholesterol is indicated by a black dashed box. **e** Close-up view of putative binding site of cholesterol in HCAR2. HCAR2 is shown in surface representation. The conserved residues forming putative cholesterol binding site are shown as cyan stick models and labeled.

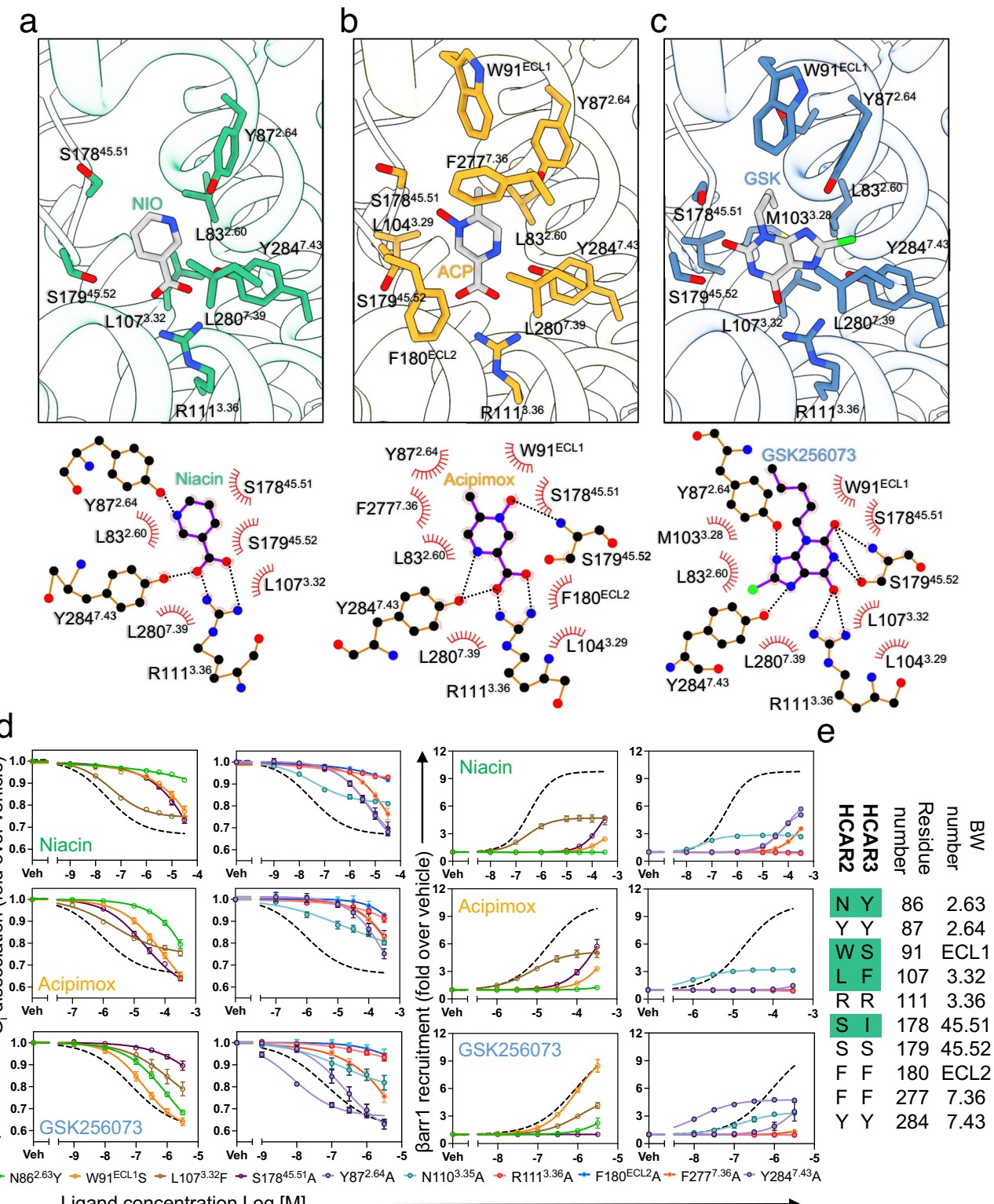

**Fig. 3 | Agonist binding pocket of HCAR2.** The ligand binding pocket with (**a**) niacin, (**b**) acipimox and (**c**) GSK256073. Residues interacting with the ligand are shown as sticks (top). Schematic diagrams of the interactions between the agonists and HCAR2, drawn by Ligplot+ (bottom). Hydrogen bonds are shown as black dotted lines and the spoked arcs represent residues making non-bonded contacts with the ligand. The carbon, nitrogen, oxygen, and chlorine atoms are colored in black, blue, red, and green, respectively. Residues in HCAR2 forming hydrogen bonds with the agonist are shown as ball and stick models. Hydrogen bonding interactions are shown as dashed lines. The ligand is shown as ball and stick model. **d** The NanoBiT-based assays to measure $G_i$ dissociation and β-arrestin1 recruitment of the wild-type HCAR2 (black dashed lines) and the ligand-pocket mutants. Circles and error bars represent mean and SEM, respectively, of three independent experiments. **e** Comparison of the residues forming the ligand binding pocket of HCAR2 with their equivalents in HCAR3.

(Fig. 3a). Acipimox has a nitrogen atom closer to its carboxyl group, and instead forms a hydrogen bond with Y284$^{7.43}$ (Fig. 3b). GSK256073, a purine-derivative carrying a pentyl tail, has almost 2.5 times the size of niacin and acipimox (Fig. 3c, Supplementary Fig. 5b). Y87$^{2.64}$ forms a similar hydrogen bond to the one seen in the niacin complex, but GSK256073 is also able to engage the side-chain of S179$^{45.52}$ (Fig. 3c). Its carbonyl group interacts with R111$^{3.36}$, but apparently more weakly than the carboxyl group of the other two ligands. The larger ligand is accommodated by a general expansion of the pocket, for example W91$^{ECL1}$ moving away from the alkyl group, giving a cavity roughly twice the size of that seen in the other complexes (Fig. 3c, Supplementary Fig. 5b). Other smaller conformational differences also help to accommodate GSK256073 in the orthosteric pocket. The residues Y87$^{2.64}$ and Y284$^{7.43}$ near the chlorine are nudged out of their original positions within the pocket, while a portion of ECL2, including S178$^{45.51}$ and S179$^{45.52}$, is shifted outward by -1.2 Å. Additionally, L83$^{2.60}$, M103$^{3.28}$ and L107$^{3.32}$ change a rotamer to avoid steric hinderance with the pentyl tail of GSK256073. (Fig. 3a–c). Other interactions between HCAR2 and the small molecule ligands include hydrophobic contact with L83$^{2.60}$, L107$^{3.32}$, S178$^{45.51}$, L280$^{7.39}$ and Y284$^{7.43}$; F180$^{ECL2}$ and (to a lesser extent) F277$^{7.36}$ form stacking interactions with the ligands (Fig. 3a–c). In the NanoBiT assay, the N86$^{2.63}$Y, Y87$^{2.64}$A, W91$^{ECL1}$S, L107$^{3.32}$F, N110$^{3.35}$A, S178$^{45.51}$A, F277$^{7.36}$A and Y284$^{7.43}$A mutants showed marked decreases in G$_i$ activity and β-arrestin1 recruitment; R111$^{3.36}$A and F180$^{ECL2}$A lost G$_i$ activity and β-arrestin1 recruitment completely (Fig. 3d, Supplementary Fig. 6 and Supplementary Data 1). We confirmed unchanged surface expression levels for these mutants (Supplementary Fig. 6a and Supplementary Data 1). Some mutants, notably Y87$^{2.64}$A, W91$^{ECL1}$S and Y284$^{7.43}$A, showed a larger impact on the effects of niacin or acipimox than GSK256073 (Fig. 3d), probably due to the more extensive protein contacts with the larger ligand (Figs. 3c and Supplementary Fig. 5b). The Y284$^{7.43}$A mutant shows a slightly increased G$_i$ activity in response to GSK256073, possibly reflecting relief of the steric hindrance between this ligand and Y284$^{7.43}$ (Fig. 3d).

## Putative determinants of ligand selectivity between HCAR2 and HCAR3

HCAR2 and HCAR3 show distinct differences in certain residues lining the ligand pocket (Fig. 3e and Supplementary Fig. 7), and introducing into HCAR2 the equivalent residues of HCAR3 causes loss of function (Fig. 3d). The side-chain of N86$^{2.63}$ lies about 5 Å from bound niacin, and makes a hydrogen bond with the carbonyl of C177$^{ECL2}$. Mutations of N86$^{2.63}$ in HCAR2 to tyrosine (the equivalent residue in both HCAR1 and HCAR3) reduced G$_i$ activity and β-arrestin1 recruitment, probably by steric hindrance to ligand binding (Fig. 3d). Replacing W91$^{ECL1}$ with serine (the equivalent residue in HCAR3) reduces G$_i$ activity and β-arrestin1 recruitment, possibly by allowing greater ligand mobility, so that the W91$^{ECL1}$S mutation has a smaller impact on signaling by GSK256073 than by the smaller ligands (Fig. 3d). The half-maximal effective concentration (EC$_{50}$) of niacin for the L107$^{3.32}$F mutant is similar to that of the wild-type, but the maximum response (E$_{max}$) of this mutant is less than that of the wild-type, both for dissociation of the G$_i$ heterotrimer and also recruitment of β-arrestin1. These results imply that L107$^{3.32}$ plays a more crucial role in HCAR2 activation rather than niacin binding (Fig. 3d, Supplementary Fig. 6b, c).

The related MK-1903, a partial agonist of HCAR2 developed by Merck, passed into Phase 2 clinical trials as a treatment for atherosclerosis and dyslipidemia, but was dropped from further development because the observed elevation of HDL cholesterol did not meet the target criteria (https://drugs.ncats.io/drug/62N05GRI0P). Molecular docking of MK-1903 provides an insight into its interactions with HCAR2 (Supplementary Fig. 8a). The adjacent nitrogen atoms appear to be capable of forming hydrogen bonds with both R111$^{3.36}$ and Y284$^{7.43}$, while the apolar fused cyclopropane and cyclopentane moiety interact with W91$^{ECL1}$, S178$^{45.51}$ and F277$^{7.36}$ giving both strongly polar and apolar interactions with the protein. The molecule is reported to be a full agonist for HCAR2 but inactive against HCAR3. Although minor differences among residues on the inner face of TM2 may play a role (N86$^{2.63}$ vs. Y86$^{2.63}$ in HCAR3, for example), and W91$^{ECL1}$ is replaced with serine in HCAR3, it is possible that some selectivity is achieved through the replacement of S178$^{45.51}$ with isoleucine. Active-state HCAR3 modeling with AlphaFold2[22] suggests that isoleucine at this position in HCAR3 leads to a conformational change in ECL2 that causes a clash with MK-1903 (Supplementary Fig. 8b).

## HCAR2 ligand binding pathway

No direct ligand binding pathway from the protein exterior is seen in the static, ligand-bound structures of HCAR2. Two different access routes can be envisaged for ligands to reach the orthosteric binding site, beginning at different clusters of positively charged residues on the extracellular surface. One of these is formed by K15$^{Nterm}$, K16$^{Nterm}$, K165$^{4.63}$ and K166$^{4.64}$, and the other is near H9$^{Nterm}$/R22$^{1.27}$ (Fig. 2d). To understand the ligand entry pathway, we performed conventional MD simulations started with the ligand away from the exterior face of the protein (Supplementary Fig. 9e), and sampled spontaneous initial binding events during 1-μs MD runs. Examining the frequency of ligand interactions with different residues showed that the negatively-charged ligands frequently approached the H9$^{Nterm}$/R22$^{1.27}$ region (Fig. 4a). To test this result experimentally, we individually mutated the residues K16$^{Nterm}$, R22$^{1.27}$ and K165$^{4.63}$ to tryptophan or glutamic acid. While most mutations cause no significant change to G$_i$ signaling, mutating R22$^{1.27}$ with either tryptophan or glutamate decreased the response to ligands (Fig. 4b, Supplementary Fig. 9g). Furthermore, MD simulations of the R22$^{1.27}$W mutant were performed, and the minimal distance measured between niacin and R111$^{3.36}$ to gauge the ligand penetration into the protein interior. The distribution of the minimum distances during the MD simulations was clearly different for the wild-type and the mutant (Fig. 4d and Supplementary Fig. 9b). The MD results that the ligand entered the R22$^{1.27}$W mutant less than the wild-type were consistent with the experiments. In addition, the interaction pattern of the mutant suggested that niacin tended to stay near H9$^{Nterm}$ and W22$^{1.27}$ (Fig. 4c). This suggests that R22$^{1.27}$ lies close to the route used by niacin and other ligands to enter the binding pocket. Due to the limited timescale of the MD simulations, the ligand never reached the orthosteric site (Supplementary Fig. 9c, f). Unlike the ligand binding pose observed in the experimental structure, the ligand contacts in the encounter structures observed in MD simulations were not stable. The simulations were repeated, but the pattern of interacting residues on the extracellular side did not change between 500-ns and 1000-ns runs. (Supplementary Fig. 9a). As discussed above, the ligand entry pathway up to the encounter structure at the surface of the protein (Fig. 4d), which corresponds to the initial stage of the ligand binding pathway, could be verified in the conventional MD and the mutant experiments. However, there was a still limitation in the timescale of conventional MD to reveal the entire binding pathway from the surface to the internal orthosteric site of the protein. The introduction of enhanced sampling techniques with MD, such as the well-tempered metadynamics, which has been recently applied to the ligand binding pathway for GPCRs[23,24] or accelerated MD[25,26] might help to reveal the entire ligand binding pathway, and thereby afford better understanding of the dynamics of activation.

## Activation of HCAR2

To gain insight into the activation mechanism of HCAR2, the structure of the complex was compared to that of apo state inactive HCAR2 (PDB:7ZLY)[27], MK6892-bound HCAR2 (PDB:7XK2)[27] and the P2Y purinoceptor 1 (P2Y1) bound to an antagonist (PDB:4XNV)[28]. P2Y1 has the highest sequence homology to HCAR2 of all proteins in the PDB. In comparison to the models of inactive HCAR2 and P2Y1, the G$_i$-coupled HCAR2 structures adopt an active form, with a significant shift of the

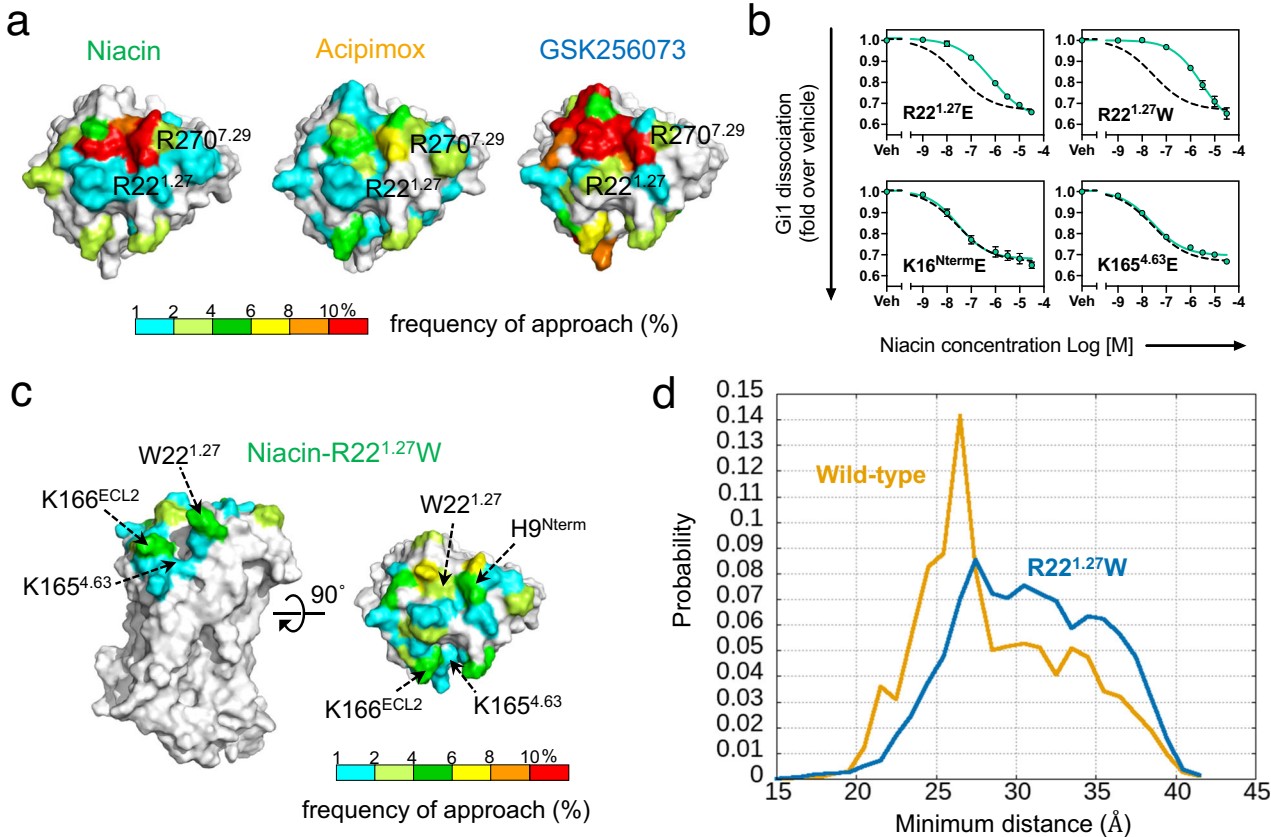

**Fig. 4 | Molecular dynamics simulation to investigate the ligand binding access.**
**a** Residues of HCAR2 contacted by the ligands, colored by frequency of contact
(niacin, left; acipimox, center, GSK256073, right). **b** The NanoBiT G-protein dis-
sociation assay of the wild-type HCAR2 (dashed lines) and the ligand-entry-pathway
mutants. Circles and error bars represent mean and SEM, respectively, of three
independent experiments. **c** Niacin contact to surface residues of the HCAR2
mutant R22[1.27]W. Molecular surfaces are colored as in (**a**). **d** Distributions of the
minimum distance between the nearest non-hydrogen atoms of niacin and R111[3.36].
The wild-type and the R22[1.27]W mutant are colored as orange and blue, respectively.
each distribution was calculated using the three independent MD runs.

extracellular end of TM2, and the marked outward movement of TM6
typical of class A GPCRs (Supplementary Fig. 10). The highly con-
served "toggle switch" residue W[6.48] of the CWxP motif found in 70%
of class A GPCRs is replaced by phenylalanine (F244[6.48]) in HCAR2.
Agonist binding causes rotamer conformational changes that are
relatively modest at the toggle switch, but which gradually increase in
the surrounding residues including the PIF, N/DPxxY and DRY motifs
(Fig. 5a-e). The outward movement of TM6 causes a vertical shift of
the toggle switch, breaking the interaction between F244[6.48] and
N110[3.35] in TM3 (Fig. 5b). In the PIF motif, F240[6.44] displays an inward
shift of P200[5.50] (Fig. 5c), which is typical of class A GPCR
activation[29,30], suggesting that these residues help to relay the signal
from the agonist to the cytoplasmic side of the GPCR[31-33]. Both the
CWxP and the PIF motifs transmit signal from the orthosteric binding
site through the allosteric sodium binding site and N/DPxxY motif to
the DRY motif at the G protein interface. HCAR2 is unusual in having
D290[7.49] (almost 90% of class A GPCRs have asparagine). This aspar-
tate residue, together with D73[2.50], S114[3.39], N286[7.45] and S287[7.46], form
the putative sodium ion binding site which is generally important for
activation by agonists[34]. Structural comparison shows rearrange-
ments of N286[7.45], S287[7.46] and D290[7.49] residues which may involve
displacements of water molecules and the sodium ion during the
activation of typical class A GPCRs (Fig. 5d). The side-chains of
D290[7.49] and Y294[7.53] lie close enough to form a hydrogen bond. Near
the cytoplasmic end of TM3, the conserved R125[3.50] of the DRY motif is
buried between TM6 and TM7, making no close polar interactions.

TM6 of HCAR2 presents only hydrophobic side-chains toward R125[3.50]
(Fig. 5e), and has no side-chains carrying carboxylic acid groups to
form the ionic lock associated with the inactive state of other GPCRs[35].
Conformation changes of R125[3.50] and Y294[7.53] in the DRY motif lead to
formation of an electrostatic interaction between these two residues
(Fig. 5e). HCAR2 undergoes a similar activation mechanism mediated
by the conserved motifs in class A GPCRs[36]. One interesting point is
that even among the activated structures, the conformational chan-
ges of the residues involved in the signaling cascade vary depending
on the type of ligand bound (Fig. 5b–e). This could be one of the
reasons for the differences observed in the binding interface of G
proteins and, furthermore, degrees of activation, depending on the
ligand.

## G-protein interface

Dozens of structures of $G_i$ coupled class A GPCRs have now been
reported, allowing their general features to be studied[36]. To allow
binding to the Gα protein C-terminal α5 helix, a cytoplasmic cavity
(largely lined by TM3, TM5 and TM6) is opened by movement of TM6
towards the cell interior. The HCAR2-$G_i$ complex structures show very
similar $G_i$ interfaces to other class A GPCRs. The HCAR2-$G_i$ complex is
stabilized by hydrophobic interactions between TM3 (V129[3.54]),
TM5(I211[5.61], L215[5.65]) and TM6 (I226[6.30], A229[6.33] and I233[6.37]) and apolar
side-chains of Gα$_i$, such as I344[G.H5.16] (Gα numbering is shown as
superscripts), L348[G.H5.20], L353[G.H5.25] and Phe354[G.H5.26] (Fig. 5f). ICL2 of
HCAR2 forms a short α helix, like many class A GPCRs, that also

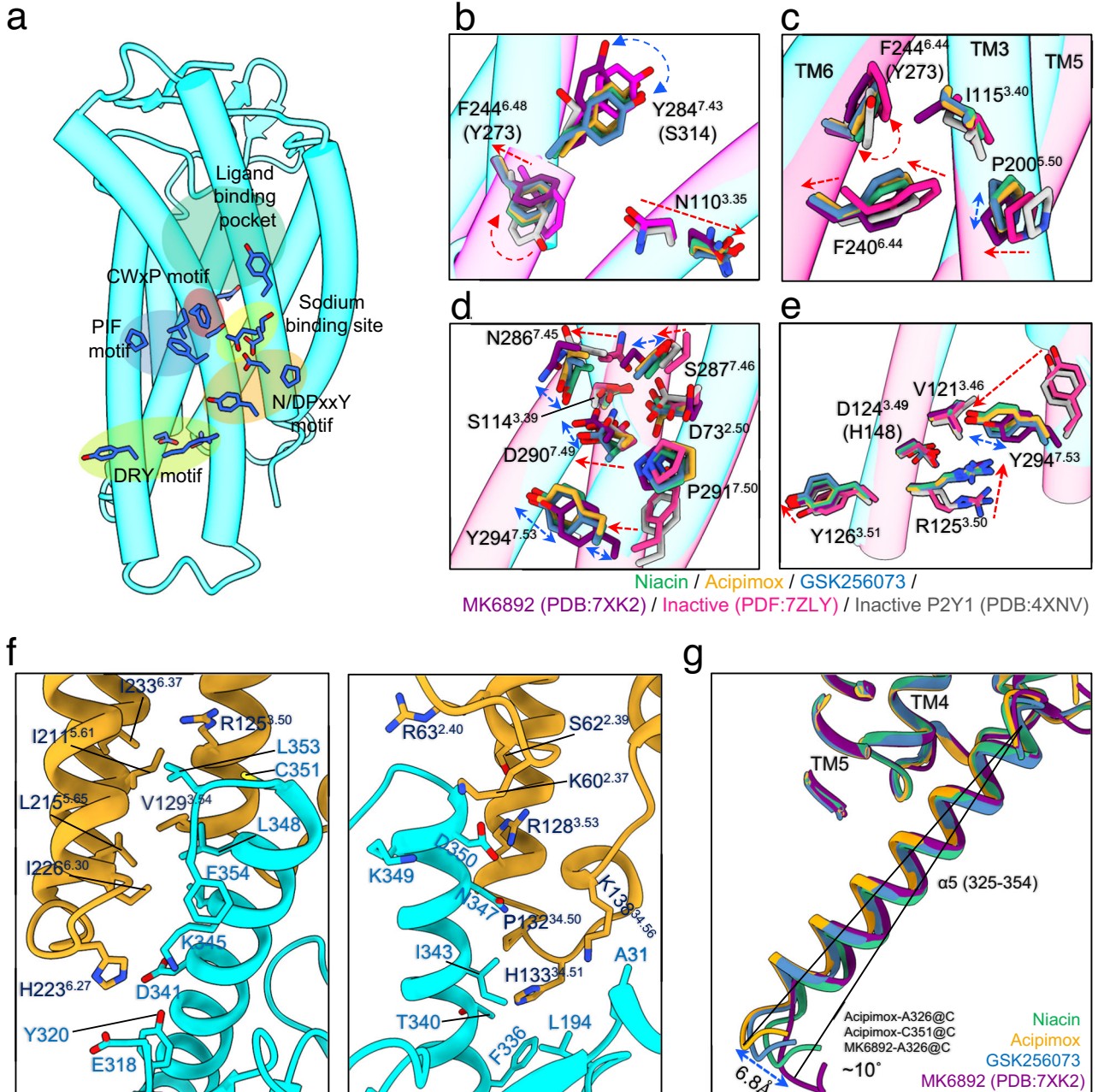

**Fig. 5 | Activation mechanism of HCAR2 and Gα$_i$ protein interactions. a** Motifs within HCAR2, shown as colored patches on the niacin bound HCAR2 structure (green, ligand binding site; red, CWxP motif; blue, PIF motif; pale yellow, sodium site; orange, NPxxY motif; yellow-green, DRY motif). **b–e** Magnified views of motifs (green, niacin-bound; orange, acipimox-bound; blue, GSK256073-bound; magenta, MK6892-bound; pink, Apo state) and the antagonist-bound inactive P2Y1 (gray). Each residue is shown as a stick model, and residues in parentheses are equivalent residues from P2Y1. The conformational changes are shown around (**b**) the CWxP motif, (**c**) the PIF motif, (**d**) the sodium site and the NPxxY motif and (**e**) the DRY

motif. The red arrows indicate conformational changes corresponding to activation, while the blue arrows indicate conformational changes based on the bound ligands. **f** The binding interface of HCAR2 (orange) with Gα$_i$ (cyan) in presence of acipimox. The residues involving in interaction are shown as stick and labeled as magenta for HCAR2, blue for Gα$_i$. (**g**) Magnified view of HCAR2 and Gα$_i$ interface. Relative position of α5 helix of Gα$_i$ is shown. Niacin bound (green), acipimox bound (orange), GSK256073 bound (blue) and MK6892 bound (magenta) HCAR2. The atoms used for calculating the angle between the α5 helices of Gα$_i$ (black line) are indicated.

contacts the Gα subunit (Fig. 5f and Supplementary Fig. 10). The principal polar interactions between HCAR2 and Gα$_i$ all lie near the cytoplasmic side of TM2, ICL2, TM3 and TM6. In particular, R128$^{3.53}$ forms salt bridge with D350$^{G.H5.22}$ and N347$^{G.H5.19}$ (Fig. 5f). This arginine is unusual, since most class A GPCRs have alanine at this position; Melatonin receptor type 1 A (MT1), which is highly specific for G$_i$ over G$_s$, has tyrosine instead. The Y$^{3.53}$ in MT1 contributes to establishing the unique conformation of G$_i$ in MT1-G$_i$ complex structure by forcing the α5 helix away from TM5[37,38]. However, R128$^{3.53}$ is much more flexible

than tyrosine and can avoid steric clash (Supplementary Fig. 11a). Although three HCAR2 structures have nearly identical interactions with the G$_i$ heterotrimer, subtle differences are found in the models reported here (Fig. 5g, Supplementary Fig. 11b), which may reflect the binding of different agonists and allow different degrees of activation.

## Discussion
In this work, we determined the structures of niacin, acipimox and GSK256073-bound HCAR2 coupled to the G$_i$ heterotrimer, and

investigated the residues involved in agonist selectivity and receptor activation. Structural comparison with the model of inactive HCAR2 gives insights into the activation mechanism of HCAR2 upon ligand binding. Overall, HCAR2 preserves key residues of the activation mechanism common to its class, but notably lacks the ionic lock residues, which may assist activation. Comparison of the three different ligand-bound HCAR2 structures hints at ligand specific changes in the conserved motifs in HCAR2 during receptor activation. Earlier work has indicated a relationship between flushing and β-arrestin1 recruitment in HCAR2[14]. However, in our cellular assay, the non-flushing agonist[8] GSK256073 and niacin each gave similar levels of β-arrestin1 recruitment. Likewise, it was found by Yang and colleagues that MK-6892 produces a strong β-arrestin1 response yet exhibits weak flushing[27], leaving the mechanism underlying flushing still unexplained. With the HCAR2 Y284[7.43]A mutant, increased $G_i$ activity and decreased β-arrestin1 recruitment are only observed in the presence of GSK256073, and not with the smaller ligands. The equivalent tyrosine residue in the chemokine receptor CCR1 ($Y^{7.43}$) has recently been reported to play a 'toggle switch' role, allowing signaling biased towards either the G-protein or β-arrestin1 downstream pathway, and the size of the ligand binding to CCR1 affects signal bias[39]. Biased signaling of GPCRs can occur not only through orthosteric agonists but also by allosteric modulators[40,41]. In a recently published study, the allosteric modulator of HCAR2 Compound 9n[42] showed a $G_i$ protein-biased allosteric effect, and specific residues involved were identified[43].

Although HCAR2 shares 95% of its sequence with HCAR3, these two receptors have distinct ligand preferences. Residues lining the deeply buried orthosteric binding pocket of the models provide structural insights into the ligand selectivity of the HCARs, and explain the different preferences. Disulfide bonds in the N-terminus and extracellular loops may lower association and dissociation rates by reducing flexibility, and differences in these loops may also affect biological function among the subfamily; P168[ECL2] of HCAR2, for example, is replaced by leucine in HCAR1 and HCAR3. The three human proteins in this family each possess some unique residues forming the orthosteric binding site, and these residues, such as W91[ECL1] and S178[45,51], could be exploited to design ligands specific for particular HCAR subtypes (Fig. 2b–d and Supplementary Fig. 5a). Since HCAR2 is of interest not only for the purpose of controlling serum lipid levels, but also inflammation and immune response, we hope that our molecular models and associated studies help the development of novel ligands with high selectivity for different members of the subfamily, and to elucidate the complex biology of HCAR2 and its signaling pathways.

## Methods

### Expression and purification of HCAR2

A coding sequence was designed to express full-length human HCAR2 (UniProtKB-Q8TDS4) carrying thermostabilized apocytochrome b562RIL (BRIL) at the N-terminus. The DNA sequence was codon optimized for expression in *Spodoptera frugiperda* (Sf9; Thermo Fisher Scientific, A35243) cells and synthesized by Integrated DNA Technologies. The gene was cloned into a modified pFastBac HT-B vector containing N-terminal haemagglutinin (HA) signal sequence, Flag-tag (DYKDDDD), 10 × His tag and HRC3V protease cleavage site. HCAR2 was expressed in Sf9 insect cells using a Bac-to-bac expression system. Cells were infected at a density of $2 \times 10^6$ cells per ml and incubated at 27℃ for 60–72 h till harvest. To purify membrane fractions, harvested cells were disrupted in a hypotonic buffer (10 mM HEPES pH 7.5, 10 mM $MgCl_2$, and 20 mM KCl) and subsequently in a high osmotic buffer (10 mM HEPES pH 7.5, 1 M NaCl, 10 mM $MgCl_2$, and 20 mM KCl) in the presence of protease inhibitor cocktail (Roche). Purified membranes were solubilized at 4℃ for 2 h in a buffer containing 50 mM HEPES pH 8.0, 500 mM NaCl, 10% glycerol, 1% (w/v) n-dodecyl-beta-D-maltopyranoside (DDM, Anatrace), 0.2% (w/v) cholesterol hemisuccinate (CHS, Anatrace) with 100 μM of ligands. Insoluble debris was removed by ultracentrifugation in 264,902 *g* at 4℃ for 40 min, and the supernatant was incubated with TALON® metal affinity resin (Clontech) overnight at 4℃. The resin was washed three times with 10 column volumes washing buffer I containing 50 mM HEPES pH 8.0, 500 mM NaCl, 10% glycerol, 20 mM IMD, 0.05% DDM, 0.01% CHS, 100 μM ligand and then washing buffer II containing 50 mM HEPES pH 8.0, 500 mM NaCl, 10% Glycerol, 20 mM IMD, 0.05% LMNG, 0.005% CHS, 0.01% DDM, 0.002% CHS, 100 μM ligand and then another 10 column volumes of washing buffer III containing 50 mM HEPES pH 8.0, 500 mM NaCl, 10% Glycerol, 20 mM IMD, 0.05% LMNG, 0.005% CHS, 100 μM ligand. The protein was then eluted using the washing buffer III supplemented with 350 mM imidazole. Eluted protein was concentrated and subjected to PD-10 Desalting Column (Cytiva). To cleave the Flag-tag and 10 × His tag, HRV3C protease was treated to purified protein. After reverse column work, fractions containing HCAR2 were pooled and subjected to size-exclusion chromatography on a Superdex 200 Increase 10/300 GL column (Cytiva) pre-equilibrated in 50 mM HEPES pH 8.0, 100 mM NaCl, 1 mM MgCl2, 0.5 mM TCEP, 0.05% LMNG, 0.005% CHS in presence 100 μM ligand. The monomeric fractions were collected and concentrated for receptor-G protein complex formation.

### Expression and purification of heterotrimeric G-protein and scFv16

The construct for heterotrimeric $G\alpha_{i1}$, $G\beta_1$ and $G\gamma_2$ was designed and purified as the same scheme as the previous report[32]. Briefly, $G\gamma_2$-(GSA)$_3$-$G\alpha_{i1}$ was inserted in downstream of P10 promoter and $G\beta_1$ was inserted downstream of AcMNPV polyhedrin promoter of pFastbac-dual vector. The scFv16 gene was cloned into a modified pFastBac1 vector with N-terminal GP67 signal sequence and C-terminal HRV3C protease cleavage site followed by 6×His-tag and purified as previously reported[32] with simple modification. In brief, media containing secreted scFv16 was separated by centrifugation at 72–96 h after infection. The pH of the medium was adjusted to 7.5–8.0 by adding Tris-base powder. Chelating agents were quenched by addition of 1 mM nickel chloride and 5 mM calcium chloride and stirring at room temperature for 1 h. After centrifugation, the supernatant was mixed and incubated with 5 ml of Ni-EXCEL resin (Cytiva). After 2 h, the resin was collected and washed with 20 column volumes of buffer containing 20 mM HEPES pH 7.5, 500 mM NaCl and 20 mM imidazole. The scFv16 was eluted with 20 mM HEPES pH 7.5, 100 mM NaCl and 350 mM imidazole. After HRV3C protease treatment, sample was further purified by size-exclusion chromatography using a Superdex 200 16/600 pg column (Cytiva). The peak fraction was collected and concentrated to 5 mg per ml for future use.

### Formation of HCAR2−G_i heterotrimer-scFv16 complex

HCAR2 and $G_i$ heterotrimer were mixed in a 1:1.2 molar ratio with 2.5 μl of Apyrase (NEB) and incubated at 25 °C for 30 min. Purified scFv16 was added to a 1:1.3, $G_i$ heterotrimer:scFv16 molar ratio. The mixture was incubated on ice for 1 h. Then the mixture was subjected to size-exclusion chromatography on a Superdex 200 Increase 10/300 GL column (Cytiva) pre-equilibrated with 20 mM HEPES pH 8.0, 100 mM NaCl, 1 mM MgCl2, 0.5 mM TCEP, 0.001% LMNG, 0.0001% CHS and 100 μM ligand. Peak fractions containing HCAR2−$G_i$ heterotrimer-scFv16 complex were collected and concentrated to 5 mg per ml.

### Cryo-EM grid preparation and data collection

Three μL of the sample was applied to a glow-discharged holey carbon grid (Quantifoil R1.2/1.3, Cu, 300 mesh). The grid was blotted with blotting force of 10 for 5 s at 4 °C, 100% humidity and flash-frozen into liquid ethane using Vitrobot Mark IV instrument (Thermo Fisher Scientific). After being plunge-frozen in liquid ethane, grids were stored in

liquid nitrogen and subjected to cryo-EM data collection. Cryo-EM imaging was performed on a Titan Krios G4 (Thermo Fischer Scientific) operated at 300 kV, equipped with a Gatan Quantum-LS Energy Filter (slit width 15 eV) and a Gatan K3 direct electron detector at a nominal magnification of 105,000× in Correlated Double Sampling (CDS) mode, corresponding to a pixel size of 0.83 Å per pixel for 49 frames with total dose of 51.7 e/Å$^2$, exposure time of 4.7 s, and dose on camera of 7.5 e$^{-1}$/px/s. Defocus range was −0.8 to −1.8 μm. All data were collected automatically using EPU software.

## Cryo-EM data processing

The processing of the collected data HCAR2 complex with each compound was carried out by cryoSPARC (v.3.3.1 and v.4.0.3)[44]. Motion correction was performed by Patch motion correction. CTF estimation for micrographs was performed by Patch CTF estimation. Micrographs under 5 or 10 Å CTF resolution were cut off by Curate Exposures.

For the HCAR2 complex with niacin, 5001 movies were collected. The 2,135,798 particles were automatically picked by blob picker and extracted using binning state (3.35 Å/pixel). Extracted particles were classified by 2D classification, and particles included in the classes showing good alignment were subjected to Ab-Initio Reconstruction. Hetero Refinement was conducted using all volumes from Ab-Initio Reconstruction. One class having a clear volume was selected and particles included in that class were extracted in 1.07 Å/pix. Using re-extracted particles, NU-refinement was conducted and 3.2 Å map was obtained. Three rounds of Hereto Refinement were conducted using the 3.2 Å map and four of junk volume. The particles included in one class having a well resolved volume were extracted in 1.245 Å/pix, then NU-refinement and DeepEMhancer[45] were conducted using extracted particles to obtain a 2.96 Å map. To improve further the quality of density for the receptor, a soft mask was applied to the receptor for Local Refinement. The final 3.43 Å map was used for modeling the structure.

For the HCAR2 complex with acipimox, 4301 movies were collected. Particles were automatically picked by blob picker from 500 micrographs, and extracted in binning state (3.11 Å/pixel). 2D classification was conducted to make template for Template picker. The 4,581,295 particles were picked by Template picker from all micrographs and were extracted in binning state (3.11 Å/pixel). Extracted particles were classified by 2D classification, and particles included in the classes showing good alignment were subjected to Ab-Initio Reconstruction. Two rounds of Hetero Refinement were conducted using all of volumes from Ab-Initio Reconstruction. One class having a clear volume was selected and particles included in that class were extracted in 1.245 Å/pix. Using re-extracted particles, NU-refinement was conducted and a 2.63 Å map was obtained. Subsequently, new masks were made to cover the entire complex. Relion (v.4.0.0) was then used for multibody refinement focusing on the receptor and a part of Gα$_i$. 3D classification was performed with Relion, and two classes were chosen with a clear density for the receptor. Finally, 228,127 particles were selected, and 3D reconstruction was performed with non-uniform refinement by cryoSPARC, and a 2.77 Å map was obtained. To improve further the quality of density maps of the receptor and Ligand (Acipimox), a soft mask was applied to the receptor for Local Refinement. The final 3.13 Å map was used for modeling the structure.

For the HCAR2 complex with GSK256073, 4957 movies were collected. Particles were automatically picked by blob picker from 500 micrographs, and extracted in binning state (3.11 Å/pixel). 2D classification was conducted to make template for Template picker. The 5434,103 particles were picked by Template picker from all micrographs and were extracted in binning state (3.11 Å/pixel). Extracted particles were classified by 2D classification, and particles included in the classes showing good alignment were subjected to Ab-Initio

Reconstruction. Two rounds of Hetero Refinement were conducted using all of volumes from Ab-Initio Reconstruction. One class having a clear volume was selected and particles included in that class were extracted in 1.245 Å/pix. Using re-extracted particles, NU-refinement was conducted and 3.22 Å map was obtained. Subsequently, new masks were made to cover the receptor and the complex of G-protein. Relion (v.4.0.0) was then used for multibody refinement focusing on the receptor, 3D classification was performed with Relion, and one class was chosen with clear density for the receptor. Then Hetero Refinement was conducted using the complex model from previous NU-refinement model and volumes from junk 2D images. Finally, 153,633 particles were selected, and 3D reconstruction was performed with non-uniform refinement by cryoSPARC, and a 3.39 Å map was obtained. To improve further the quality of the density map, a soft mask was applied to the receptor for Local Refinement. The final 3.74 Å map was used for modeling the structure.

## Model building and refinement

The predicted model of HCAR2 from AlphaFold Protein Structure Database was used as the initial template for modeling the receptor. Model of G-proteins and scFv16 complexes was used from the structure of Type 2 bradykinin receptor-Gq complex (PDB:7F2O)[46]. At first, the initial models were roughly fitted to each cryo-EM map using ChimeraX[47]. Fitted models were subjected to real-space refinement in PHENIX[48]. The refined model was manually adjusted using COOT[49] and then further refined with the Real-space refinement procedure in PHENIX[48]. ChimeraX[47], LigPlot+[50] and CASTp 3.0[51] were used to prepare figures illustrating structural information in the paper.

## NanoBiT-G-protein dissociation assay

HCAR2 ligand-induced G$_i$ activation was measured by the NanoBiT-based G-protein dissociation assay[20], in which the interaction between a Gα subunit and a Gβγ subunit was monitored by the NanoBiT system (Promega). Specifically, a NanoBiT-G$_{i1}$ protein consisting of Gα$_{i1}$ subunit fused with a large fragment (LgBiT) at the α-helical domain (between the residues 91 and 92 of Gα$_{i1}$) and an N-terminally small fragment (SmBiT)-fused Gγ$_2$ subunit with a C68S mutation was expressed along with untagged Gβ$_1$ subunit and HCAR2 (containing the N-terminal HA-derived signal sequence followed by the FLAG-epitope tag; ssHA-FLAG-HCAR2). The mutants of HCAR2 were generated using primers listed in Supplementary Table 3. HEK293A (Thermo Fisher Scientific, R70507) cells were seeded in a 6-cm culture dish at a concentration of $2 \times 10^5$ cells ml$^{-1}$ (4 ml per well in DMEM (Nissui) supplemented with 10% fetal bovine serum (Gibco), glutamine, penicillin and streptomycin), 1 day before transfection. Transfection solution was prepared by combining 12 μL (per dish hereafter) of polyethylenimine (PEI) Max solution (1 mg ml$^{-1}$; Polysciences), 400 μL of Opti-MEM (Thermo Fisher Scientific) and a plasmid mixture consisting of 400 ng ssHA-FLAG-HCAR2 (or an empty plasmid for mock transfection), 200 ng LgBiT-containing Gα$_{i1}$ subunit, 1 μg Gβ$_1$ subunit and 1 μg SmBiT-fused Gγ$_2$ subunit (C68S). After incubation for 1 day, the transfected cells were harvested with 0.5 mM EDTA-containing Dulbecco's PBS, centrifuged and suspended in 2 ml of HBSS containing 0.01% bovine serum albumin (BSA; fatty acid-free grade; SERVA) and 5 mM HEPES (pH 7.4) (assay buffer). The cell suspension was dispensed in a white 96-well plate at a volume of 80 μL per well and loaded with 20 μL of 50 μM coelenterazine (Carbosynth) diluted in the assay buffer. After a 2 h incubation at room temperature, the plate was measured for baseline luminescence (SpectraMax L, Molecular Devices) and titrated concentrations of niacin (20 μL; 6X of final concentrations) were manually added. The plate was immediately read at room temperature for the following 10 min, at measurement intervals of 20 s. The luminescence counts from 8 min to 10 min after ligand addition were averaged and normalized to the initial count. The fold-change values were further normalized to those of vesicle-treated samples and used

to plot the G-protein dissociation response. Using the Prism 9 software (GraphPad Prism), the G-protein dissociation signals were fitted to a four-parameter sigmoidal concentration-response curve with a constrain of the *HillSlope* to absolute values <2. For each replicate experiment, the parameters *Span* (= *Top* − *Bottom*) and $pEC_{50}$ (negative logarithmic values of $EC_{50}$ values) of individual HCAR2 mutants were normalized to those of WT HCAR2 performed in parallel and the resulting $E_{max}$ values were used to calculate ligand response activity of the mutants.

## NanoBiT-β-arrestin recruitment assay

HCAR2 ligand-induced β-arrestin1 recruitment to the receptor was measured by the NanoBiT-based β-arrestin recruitment assay. Briefly, the N-terminally LgBiT-fused human β-arrestin1 (200 ng plasmid per 6-cm dish) was expressed with the C-terminally SmBiT-fused ssHA-FLAG-HCAR2 (ssHA-FLAG-HCAR2-SmBiT; 1 μg plasmid per 6-cm dish) in HEK293A cells in a 6-cm culture dish. As a negative control, ssHA-FLAG-V2R-SmBiT (vasopressin V2 receptor) was used. The transfected cells were dispensed into a 96-well plate, and ligand-induced luminescent changes were measured by following the same procedures as described for the NanoBiT-G-protein dissociation assay, except that luminescence counts were taken from 13 min to 15 min after compound addition.

## Flow cytometry analysis

Transfection was performed according to the same procedure as described in the "NanoBiT-G-protein dissociation assay" section. One day after transfection, the cells were collected by adding 200 μl of 0.53 mM EDTA-containing Dulbecco's PBS (D-PBS), followed by 200 μl of 5 mM HEPES (pH 7.4)-containing Hank's Balanced Salt Solution (HBSS). The cell suspension was transferred to a 96-well V-bottom plate in duplicate and fluorescently labeled with an anti-FLAG epitope (DYKDDDDK) tag monoclonal antibody (Clone 1E6, FujiFilm Wako Pure Chemicals; 10 μg per ml diluted in 2% goat serum- and 2 mM EDTA-containing D-PBS (blocking buffer)) and a goat anti-mouse IgG secondary antibody conjugated with Alexa Fluor 488 (Thermo Fisher Scientific, 10 μg per ml diluted in the blocking buffer). After washing with D-PBS, the cells were resuspended in 200 μl of 2 mM EDTA-containing-D-PBS and filtered through a 40-μm filter. The fluorescent intensity of single cells was quantified by an EC800 flow cytometer equipped with a 488 nm laser (Sony). The fluorescent signal derived from Alexa Fluor 488 was recorded in an FL1 channel, and the flow cytometry data were analyzed with the FlowJo software (FlowJo). Live cells were gated with a forward scatter (FS-Peak-Lin) cutoff at the 390 setting, with a gain value of 1.7. Values of mean fluorescence intensity (MFI) from ~20,000 cells per sample were used for analysis. For each experiment, we normalized an MFI value of the mutants by that of WT performed in parallel and denoted relative levels.

## Molecular dynamics simulations

First, initial models of HCAR2 by placing the ligand 30 Å away from that at the orthosteric site toward the z-direction were prepared from the experimental complex structures for the three ligands solved by cryo-EM. Then, in the niacin and GSK256073 systems, the HCAR2 structure was replaced to that of the acipimox system, because the resolution of HCAR2 bound to acipimox was the highest among structures. G proteins bound to HCAR2 were removed, and molecular dynamics (MD) simulations were conducted using HCAR2 (residues 9–316) and the ligand. A membrane-water system was prepared using the Membrane Builder implemented in CHARMM-GUI[52–54]. Missing hydrogen atoms were added. The protonation states of the protein, charged residues and histidine residues at pH 7 were set according to the PROPKA estimation implemented in PDB2PQR:[55] D73 and E37 were set to a protonated aspartic and glutamic acid, respectively. H134 was set to the neutral histidine with only the protonated Nε atom (HSE in the

CHARMM force field), while other histidine residues were set to the neutral histidine with only the protonated $N_\delta$ atom (HSD in the CHARMM force field). The N- and C-termini were set to $NH_3^+$ and $COO^-$, respectively. The protonation state of ligands was assigned using the Epik module in Maestro (Schrödinger release 2020-4, Supplementary Fig. 9d). The protein was placed at the center of a rectangular unit cell and embedded in about 70 Å × 70 Å 1-palmitoyl-2-oleoyl-glycero-3-phosphocholine (POPC) bilayer at the x-y plane. The number of POPC molecules were ~54 and ~53 at the upper and lower leaflets, respectively. The orientation of HCAR2 relative to the membrane was set to be the same as that of succinate receptor 1 (SUCNR1) deposited in the orientations of proteins in membranes (OPM) database (ID: 6RNK), because HCAR2 and SUCNR1 have high sequence homology and similar three-dimensional structures. The ligand was initially displaced 30 Å along the z-axis direction from its position in the experimental model. After filling the unit cell with water molecules (TIP3 water model), potassium and chloride counterions were added to give a final ion density of 150 mM.

All-atom conventional MD simulations were conducted using the MD program package GROMACS ver. 2019.6[56] with PLUMED ver. 2.7.1[57]. Simulations were performed under periodic boundary conditions. The CHARMM36m force field[58–60] with WYF parameter[61] for cation-pi interactions was used for all components except the ligand. The CHARMM General Force Field (CgenFF)[62] was used for niacin. An energy minimization and six equilibration runs were performed according to the default setting in the Membrane Builder: Simulation setups were the same as those of our previous simulations[63]. After equilibration runs, production runs were performed. Due to the periodic boundary condition, the ligand at the dissociated position sometimes moved across the periodic boundary of the z-direction and reached to the intracellular side (i.e., the upper direction of the unit cell shown in Supplementary Fig. 8e). Therefore, to sample ligand binding events at the extracellular side of the protein without the artifact of the periodic boundary condition, a harmonic flat-bottom restraint was imposed on the distance between the protein and the ligand so that the ligand did not move toward the intracellular side across the periodic boundary: The UPPER_WALLS restraint in PULMED was set to the distance between Cα atom of R111 and a nitrogen atom of niacin (Supplementary Fig. 8d), and the upper distance and the force constant were set to 40 Å and 0.239 kcal/mol·Å² (i.e., 100 kJ/nm²). Here, the restraint does not bias the specific binding pathway, and the ligand spontaneously approached the extracellular surface of HCAR2 during production runs. In each ligand, three 1-μs production runs were performed independently. The ensemble was NPT ensemble, and the temperature and pressure were set at 300 K and 1 atm, respectively. The thermostat was using the Nose-Hoover scheme. The barostat was using the semi-isotropic Parrinello-Rahman approach. The electrostatic interactions were handled by the smooth particle mesh Ewald method. The van der Waals interactions were smoothly truncated using the switching function within a range of 10–12 Å. Bond lengths involving hydrogen atoms were constrained by the P-LINKS algorithm, and the time step was set to 2 fs. From each 1-μs production run, snapshots were extracted every 1 ns and used for trajectory analysis. The contact was defined as <4 Å distance between nonhydrogen atoms on the protein residues and the ligand. In addition to the wild-type, MD simulations for the $R22^{1.27}W$ mutant were performed. The $R22^{1.27}W$ substitution was done using the setup procedure of the Membrane Builder implemented in CHRARMM-GUI[52–54], and other setups and conditions of MD simulations were performed as the same for the wild-type as described above.

## Docking simulation of MK-1903

A docking simulation of MK-1903 to the HCAR2 structure was performed using Glide (Schrödinger Release 2020-4)[64]. The receptor structure was prepared by the experimental structure of the acipimox

bound HCAR2 complex solved by cryo-EM. Hydrogen atoms were added by the protein preparation wizard in Maestro (Schrödinger Release 2020-4)[65]. After an optimization and an energy minimization of the receptor structure, a cubic grid was determined with the Acipimox bound at the orthosteric site as the center. The grid size was the default size, which contained enough residues to interact with the ligand, and the grid potential parameter of the receptor was calculated without the ligand. The structure of MK-1903 was obtained from the PubChem database (CID 49763030)[56]. The protonation state of the ligand was determined using the LigPrep wizard in Maestro (Schrödinger Release 2020-4). Then, the prepared MK-1903 was docked into the grid in HCAR2 using Glide with the standard precision mode. A docking pose of MK-1903 was selected by the best docking score and shown in Supplementary Fig. 8.

## Statistics and reproducibility
Each NanoBiT-based assay was performed with at least three independent replicates, and the data were analyzed using GraphPad Prism 9 (GraphPad Software). The data are presented as means ± standard error of the mean.

## Reporting summary
Further information on research design is available in the Nature Portfolio Reporting Summary linked to this article.

## Data availability
The cryo-EM density maps and coordinates have been deposited in the Electron Microscopy Data Bank (EMDB) and the Protein Data Bank (PDB) under accession number EMD-34437, 8H2G (niacin bound HCAR2-G$_i$), EMD-36900 and 8K5B (niacin bound HCAR2 local), EMD-35234, 8I7V (acipimox bound HCAR2-G$_i$), EMD-36901 and 8K5C (acipimox bound HCAR2 local), EMD-35235, 8I7W (GSK256073 bound HCAR2-G$_i$), EMD-36902 and 8K5D (GSK256073 bound HCAR2 local) respectively. The previously published PDB coordinates used in this paper are available: 4XNV, 6RNK, 7VGZ, 7XK2, 7Y89, 7ZLY. Source data are provided with this paper.

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

## Acknowledgements

The cryo-EM experiments were performed at the cryo-EM facility of the RIKEN Center for Biosystems Dynamics Research (Yokohama). We would like to thank Prof. Tomohiro Nishizawa and Dr. Haruhiko Ehara for their help with cryo-EM data collection. We also thank Kayo Sato, Shigeko Nakano and Ayumi Inoue at Tohoku University for their assistance in plasmid preparation and the cell-based GPCR assays. This work was supported by Japan Agency for Medical Research and Development (AMED) under grant numbers JP21fk0310103 (S-Y.P.), JP20gm0010004 (A.I.), JP22fk0310517(M.I.), JP22ama121023 (M.I.), JP20am0101095 (A.I.) and JP22zf0127007 (A.I.); JSPS/MEXT under KAKENHI grant numbers JP19H05779 (S-Y.P.), JP21H02449 (S-Y.P.), JP21H04791 (A.I.), JP21H05113 (A.I.), JPJSBP120213501 (A.I.) and JPJSBP120218801 (A.I.); Japan Science and Technology Agency (JST) grants JPMJFR215T (A.I.) and JPMJMS2023 (A.I.); the Program for Promoting Research on the Supercomputer Fugaku (MD-driven Precision Medicine) (project ID: hp200129, hp210172 to M.I.), and by a Grant-in-Aid for Scientific Research on Innovative Areas Molecular Engine (18H05426 to M.I.), and by the grant for 2021-2023 Strategic Research Promotion (No. SK202202 to M.I.) of Yokohama City University. And this work was supported by the National Research Foundation of Korea (NRF) under grant numbers 2022R1A2C1011793 (Y.-H.L.), 2020R1A6C101A188 (D.-S.L.), National Research Council of Science & Technology (NST) grant number CCL22061-100 (Y.-H.L.) and KBSI grant numbers C320000, C330130, C390000, and C318410 (Y.-H.L.) by the Korean government.

## Author contributions

J.-H.P., N.I., M.O., D.-S.L. and Y.-H. L. expressed and purified proteins. J.-H.P., N.I., J.R.H.T. and S.-Y.P. collected and analyzed the cryo-EM data and refined the structure. J.-H.P., N.I., S.-Y.P., K.K., T.I. and A.I. designed the mutants. K.K., T.I. and A.I. performed and analyzed the cell-based mutant assays. T.E and M.I. performance of molecular dynamics simulation. S.-Y.P. and A.I. initiated the project, planned, analyzed experiments, supervised the research. J.-H.P., J.R.H.T., A.I. and S.-Y.P. wrote the manuscript with input from all co-authors.

## Competing interests

The authors declare no competing interests.
