## [Peer Review File · Nature Communications]

Reviewers' Comments:

Reviewer #1:

Remarks to the Author:

The authors of this paper presented high-resolution cryo-EM structures of active state HCAR2-Gi complexes with three different ligands, and performed mutagenesis and functional studies to link structural observations to ligand selectivity, receptor activation and signaling. Molecular dynamics stimulation experiments are used to discuss how the ligand enters the receptor.

However, I have several concerns as listed below, especially with the consideration of the recently published HCAR2 structures by another group (although I have no doubt with the independency of the work presented in this manuscript), I do not think this manuscript is suited for publication in Nature Communications.

1. Another work on HCAR2 structures, including inactive and active conformations, was reported recently (Nat Commun 14, 1692 (2023)). Although it does not affect the independency of the work presented in this manuscript, people may expect some new advancements compared to the published work.

2. It would be better to include a sequence comparison between HCAR2 and HCAR3, as well as with P2Y1 in the region critical for activation, to make the conclusion more reliable. Additionally, I'm not sure if P2Y1 is a good choice for the inactive structure, some other reported structures from the δ branch of Class A GPCR or the recently published crystal structure of HCAR2 might be considered.

3. In line 116 and Supplementary Fig.4, the authors mention that "the extracellular face shows a distinct pattern of electrostatic". It is unclear if this phenomenon is specific to HACRs and whether this feature has any special impact on its functionality.

4. The Results session of "Orthosteric binding site" is quite long and may need to be better organized. For example, the comparison of HCAR2 with HCAR1/HCAR3 and the comparison between the three different ligands could be separated.

5. There are mutagenesis experiments in this manuscript, but the expression levels of the mutated constructs are lacking. It would be beneficial to include data on the key mutations to demonstrate that the mutations do not significantly affect expression levels. Without this information, the conclusions may not be convincing.

6. Line 166-167, "suggesting that this tyrosine and TM7...controlling the balance of ..." this conclusion is quite strong, but I am not fully convinced by how this conclusion is supported. Especially the authors made an analogy with the case in CCR1, which I find quite confusing.

7. The Molecular dynamics stimulation experiments were used to study the ligand entry, which is an interesting direction but certainly needs more experimental support.

8. The Discussion session is too simple, and seems to be written in a rush. There are some interesting directions that could be discussed, for example, the ligand entry path, the selectivity, and signaling bias. What have been found through this study, which can be further investigated.

9. Line 293, the last sentence in the Discussion mentioned the side effect of flushing, however, it is not clear how this side effect is caused, is it due to lack of selectivity between different HCARs, or b-arrestin signaling, or other pathways? Without understanding this, I'm not convinced that this study could provide insight into design of drugs with reduced side effect.

10. In Supplementary Fig.1, the band close to HCAR2 is labeled with Gai1-G γ 2, is it a fusion expression? I didn't see this in the Methods.

11. In Supplementary Fig.2, the text overlaps each other, maybe due to the conversion from word to PDF.

12. The writing of this manuscript needs more work to polish. A lot of typos and grammatical problems. Just to give a few examples: Line 261, "in presence niacin and acipimox"; Line 265-266, "but R128...has not such contribution", this sentence is confusing.

Reviewer #2:

Remarks to the Author:

The manuscript describes 3 new high-resolution cryo-EM structures of agonist-bound human hydroxycarboxylic acid receptor type 2 (HCAR2) in the active state bound to inhibitory G protein (Gi). The authors observed similarity of receptor - G protein structure to those of other A-class GPCRs. They clearly observed orthosteric ligand binding pockets with ligand structures and observed some ligand-induced changes especially evident for a larger ligand, GSK256073. They performed a series of HCAR2 mutations to investigate its ligand binding pathways, pocket and G protein binding, which they tested using NanoBiT protein interaction assays focused on beta-arrestin binding and G protein dissociation. Endogenous agonist niacin binding was also tested via atomistic molecular dynamics (MD) simulations with the results on ligand binding pathways in agreement with NanoBiT experimental assays.

This is an important study, which can be used for development of novel selective HCAR ligands, which can be used for treatment of dyslipidemia, inflammation and a variety of other disorders. The manuscript is written fairly well and is concise and well-focused. There are no major language issues with some minor grammar problems in part indicated below. The combination of experimental and computational techniques corroborating results of each other is also a big plus of this study,

There are some issues, however, the authors need to address before the manuscript can be published.

1. The manuscript can be strengthened by adding a more comprehensive Discussion section, which can be focused on comparing results of this study with previous ones on similar systems, i.e. homologous GPCR in terms of structures and previous molecular/cell biology experiments on HCARs, for example. Current version of the "Discussion" section just provides some brief general overview of the study results. Also, there are some discussion elements embedded in the Results section, e.g. on lines 158-167 and 180-184.

2. The description of NanoBiT assays and their results are somewhat confusing. Sometimes in the manuscript their results are referred to as "signaling" and sometimes as "binding" and "dissociation." This along with what quantities are being measured need to be clarified. It would be great to report some quantities such as K_d , K_i or equivalent for easier comparison than plots if this is amenable. I was also surprised that ligand binding affinities or other similar quantities were not even mentioned in the study because they represent very important quantities for comparing different ligands and/or protein mutants. Another important consideration is comparison of beta-arrestin and G protein assays, which the authors only glanced over in the manuscript. However, this is a very important subject alluding to biased signaling, which can be analyzed and discussed much more (see comment 1 above).

3. There is no amino acid sequence information in the supporting text for HCAR2 and other related GPCR discussed. Sequence alignment and comparison will be very useful.

4. The description of molecular dynamics (MD) simulations and their analyses are somewhat confusing. It is not clear what kind of simulations were performed and whether convergence was achieved. It seems that they were largely unbiased runs with flat-bottom z position restraints used to prevent ligand flipping over to the intracellular side. Does it mean that spontaneous ligand binding was probed? Did a ligand ever reach an orthosteric site during microsecond-long MD runs? It is not clear from Figure 4D, which does not provide many useful insights. If a main purpose was sampling of initial extracellular binding as shown in Fig. 4A&C, this needs to be clarified as well. Also, some quantitative analysis using RMSD from the orthosteric binding poses or interaction energies using e.g. MM-PBSA or similar approach could be potentially used.

5. Molecular details of HCAR2 - Gi interactions as shown in Fig. 5F and G remain very unclear. It would be best to show specific inter-residue interactions e.g. in Fig. 5G, at least the most important ones. So, Gi alpha5 residues can be also schematically shown and their H bonding or other directional interactions with the receptor residues indicated e.g. by thin dashed lines.

6. Molecular docking of MK-1903 shown in Supplemental Fig. 6 is not very clear in terms of purpose, methodology etc. Docking methods should be briefly mentioned. More importantly, it is not clear what is being compared. If it is selectivity against HCAR3 as indicated on lines 189-190 of main text, then docking to homology or de novo (e.g., AlphaFold) model of HCAR3 would have also needed to be done. Otherwise, it is hard to conclude that "it seems highly likely that selectivity is achieved through the replacement of S178ECL2 with isoleucine. Isoleucine in this position cannot adopt a rotamer that does not strongly clash with MK-1903" Also, a comparison of AlphaFold model of HCAR2, used as a template for density fitting, with solved cryo-EM structures would be interesting to show as well.

7. A lot of references are not listed or missing, especially for the Methods section. Please check.

8. There are a number of minor issues as detailed below as well (with some overlap / duplications with ones above possible):

l 94 cryo-electron microscopy -> cryogenic electron microscopy

l 100 "dissociation and recruitment of the Gi heterotrimer (Gai1, Gβ1 and Gγ2) and β-arrestin1 (Figure 1a)" Could you please clarify if dissociation and recruitment of both Gi and arrestin were probed, as it seems from the Figure caption the dissociation of G protein and recruitment of arrestin was studied.

l 101 scFv16 - please spell out

l 132-133 "principally a hydrogen bond between R1113.36 and the ligand carboxyl group" - Is this actually a salt bridge interaction?

l 136 "pentane tail" -> "pentyl tail"

l 144 "alkane group" -> "alkyl group"

l 148 "form aromatic ring interactions with the ligands" - please specify type of interactions

l 150-151 "mutants showed marked decreases in Gi and β-arrestin1 signaling" - is it signaling or binding assay? The same is true for a few instances of "signaling" below. Then later the "dissociation" term is being used.

l 155 "a slightly heightened" - increased or decreased?

l 177 "The mutant L1073.32F exhibited similar effective concentration but decreased maximum effective compared with wild-type" - this is confusing as it is not clear what is being compared

l 191 "N862.63Y" -> "N862.63 vs Y? in HCAR3"

l 200 "ligand pathway" -> "ligand binding pathway"

l 220 "P2Y1 has the highest sequence similarity to HCAR2 of all proteins in the PDB" - what is the similarity?

l 240 "The side-chains of D2907.49 and Y2947.53 lie close enough to form a hydrogen241 bond." in the active HCAR2?

l 256 "such as I344G.H5.16" please indicate the source of Gi residue notation

l 257 "Supplementary Fig. 9a" -> "Supplementary Fig. 9a&b"

l 259 "Supplementary Fig. 8" -> "Supplementary Fig. 9b"

l 265 "complex structure by restricting the α5 helix from moving away from TM5". TM5 was not mentioned as having direct contacts with Gi alpha5

l 266 "has not such contribution" -> "does not have such contribution"

l 271 "signal bias" -> "biased signaling"

l 307-310 "Structure features" -> "Structural features" Please indicate that in panels A and B proline residues are colored in magenta

l 313-314 "N-terminus" -> "N-terminal", "C-terminus" -> "C-terminal"

l 315-316 "Hydrogen bond are shown as dotted blue lines" "Hydrogen bond are shown as dashed light-blue lines" What do distances 3.1 and 3.5 Å represent? In Fig 2E Phe residue adjacent to Q187 is shown but not labeled

l 321-322 "Residues close to the ligand" - how was the closeness defined?

l 324 "chloride atom" -> "chlorine atom". Cl is shown as green in the 3D diagram but as pink in

the schematic one, please correct

I 326-328 Comparison of dissociation constants (Kd) or equivalent numerical values in Fig. 3D if possible might be more helpful than comparison of dissociation/association plots

I 330 Residue numbers in panel E would be helpful as some residue positions as written, e.g. ones in ECL-s, are not well defined.

I 334. Residue labels in panel A are really small and hard to see.

I 334-335 How was a contact defined?

I 335 K165 mutant is shown in panel B, whereas K166 is labeled in panel C. Is it correct?

I 339 What is MD2?

I 339-340 X and Y axis tick marks are very small and hard to see

I 351. The interface in panel F is not shown clearly. only alpha-5 of Gi is shown well, while the receptor binding interface is obscured.

I 355 "orange, hydrophobic" -> "yellow, hydrophobic"

I 556-557 "CHARMM-GUI{Citation}{Citation}" -> "CHARMM-GUI" (please add citations)

I 561 "The protonation state of niacin was assigned using the Epik module in Maestro" Please indicate what protonation state was chosen

I 563 "POPC" -> please spell out

I 564 "The number of POPC molecules were ~54 and ~53 at the upper and lower leaflets" This is a really small number of lipid molecules for such a large protein system, please double check

I 567 "have high sequence homology" - what is the % of sequence identity or similarity?

I 570 "potassium and chloride counterions were added" - if the focus was on ligand binding / unbinding, sodium ions should have been used to mimic the extracellular environment.

I 577-578 "Simulation setups were the same as those of our previous simulations." Please add references and/or details. Please indicate simulation length and/or restraints used if any for equilibration and production MD runs. Please indicate how MD simulation convergence was monitored.

I 587 'The thermostat was the Nose-Hoover scheme. The barostat was the semi-isotropic' -> "The thermostat was using the Nose-Hoover scheme. The barostat was using the semi-isotropic"

I 587-593 Please add references for thermostat, barostat, PME, P-LINKS, PME etc.

I 643 "Construct" -> "Construction"

I 651 "in presence" -> "in the presence of"

I 679 SI Fig 6 It would be the best to color niacin differently, as it blends in with the receptor

I 697 "ICL3" -> "ICL2"

I 700 SI Fig 9 A What does coloring mean?

Reviewer #3:

Remarks to the Author:

The manuscript, "Structural basis for ligand recognition and signaling of hydroxy-carboxylic acid receptor 2" by Jae-Hyun Park et al. presents 3 cryo-EM structures of HCAR2 bound to different small molecule ligands. The cryo-EM structures are supported by mutagenesis and pharmacology data that provide insight into how the three different ligands bind to the orthosteric site of HCAR2, which is buried deep within the receptor. Molecular dynamics simulations and mutant/pharmacology data provide insight into a unique mechanism of how these ligands access the receptor. A comparison of the activation mechanism of HCAR2 and its interactions with G proteins were performed by comparison to the inactive P2Y1 receptor structure (closest homology structure to HCAR2).

Overall, the manuscript is well written and presented. A minor concern is that there isn't much mechanistic insight provided by this manuscript, other than how these three ligands bind HCAR2. The insight into how agonists access the orthosteric site and the activation mechanisms are rather speculative at this point and not supported by convincing experimental data.

The much larger and more significant concern is the quality of the cryo-EM models and cryo-EM maps. The provided PDB validation reports provide evidence that the modelling of the receptor is rather poor by comparison to what would be expected. Presumably, the authors would have checked over this report in detail and realized the structures needed improvement prior to

submission. There are a relatively large number of significant clashes, poorly modelled protein backbones, and protein sidechains all on the receptor chain. The Q-score indicates that there could be issues with resolvability of atoms around the receptor region (and parts of the G protein).

These concerns are heightened when looking at the provided experimental maps and models. For example, residue TYR284 is suggested to be in a different rotamer conformation in the GSK256073 structure (Fig 5B), however, it's unclear how the authors can make this claim, as evidenced in the provided screenshots in the pdf version of this review. The residue could in fact be in the same conformation as the other structures as there is convincing EM-density for it. The authors have instead modelled part of GSK here, which doesn't really seem plausible as the aliphatic chain would not have density in this shape and is instead likely to be unstructured/dynamic manner. Nevertheless, the quality of the map/models prevents a clear conclusion here.

Similarly, residue W188 in the GSK-bound structure looks like it is modelled into the noise of the map, rather than a different conformation as shown in Figure 2E. These are just a few examples. The extracellular regions are also rather messy and it is unclear how reliable the modelling would be in these regions. The authors have modelled large portions of the Galpha subunit into the noise of the map and in some cases there is no density present to support any protein modelling at all. These are all significant issues that need to be addressed, and the best place to start would be improving the quality of the cryo-EM maps. The sharpened maps that were provided appear to be over sharpened and the deep EM enhance maps are not very reliable around the regions of the receptor, which is the key region of the manuscript. I'd suggest the authors perform a local refinement around the receptor only, which should be easy to do given the data was processed in CryoSparc. Hopefully, these maps will be of better quality than the provided maps, which would significantly improve the modelling of the receptor. Otherwise, the authors made need to adjust the data processing strategies to try and improve quality of the maps (e.g. they could process the data in Relion, where 3D classifications and polishing can make a considerable improvements in the maps versus cryosparc). Once the maps are improved, the authors will need to spend time improving the modelling on all regions of the GPCR-G protein complex, and then spend time reinterpreting their main findings.

Other comments:

1) The authors should include a figure that shows the local resolution of their cryo-maps.

2) It's not made clear in the text why the authors performed the B-arrestin assays? I was expecting a discussion about bias or something, but in the end was just left wondering why the data was even there.

3) Figure 4 and the MD simulations. The quality of the structures does not support any of the data here, residues R22 and K16 are modelled into nothing/noise. As mentioned previously, the quality of the cryo-EM density around the extracellular region is rather poor, as is the modelling. I'm not sure how one can draw any conclusions from this region of the model, and the complementary MD data isn't very convincing. For example, in the comparison in Fig 4D with WT and R22W, the minimum distance is achieved in 1-2 simulations in a very transient manner. This whole section should be removed.

4) With regards to the activation mechanism, this of course difficult to do in the absence of an inactive state structure. I'm sure the editor's and authors are aware of the HCAR2 structures that were recently published in Nature Communications, which include inactive state structures. The authors should be comparing their work to those structures (both inactive and active). For example, are there any differences between the studies, other than the orthosteric ligands?

5) Does the inclusion of BRIL to N-terminus affect the pharmacology of the receptor? Given the position of the N-terminus over the top of the receptor it would be prudent to perform these experiments.

6) More details for Supp Fig 3 need to be provided. What maps are shown, what contour level, and was the density carved or masked and if so what was the resolution cut-off (i.e. provide enough that this figure can be recreated by others).

Screenshots of poor maps/modelling provided in PDF.

Response to reviewers

We thank the reviewers for their valuable comments, which have significantly contributed to improving the manuscript. Please find attached the revised version, which has been thoroughly revised in accordance with the referees' comments. Our responses to the comments of each reviewer can be found below.

Reviewer #1 (Remarks to the Author):

1. *Another work on HCAR2 structures, including inactive and active conformations, was reported recently (Nat Commun 14, 1692 (2023)). Although it does not affect the independency of the work presented in this manuscript, people may expect some new advancements compared to the published work.*

The recently published paper by Yang et al describes a crystal structure of the inactive form of HCAR2, and the cryo-EM structure of the MK-6892 complex. From this, models were calculated for the niacin complex. We have not only determined this complex experimentally, but also the complexes with acipimox and GSK256073 (which is nowhere mentioned by Yang and colleagues), and tested these ligands against a panel of mutant receptors, testing both Gi activity and β -arrestin1 recruitment. Our MD simulation of ligand entry into HCAR2 is entirely novel. Our paper also compares HCAR2 with its homologs, which the Yang paper does not, and we find support for the idea that Y7.43 plays a role in control of signal bias. We are slightly surprised that the reviewer seems to find independent analyses of little value once one has been published, and very surprised indeed he sees nothing new in our paper. We note the Yang paper was unpublished at the time we submitted, but have cited it in the revised version to highlight the novelty of our results.

2-1. *It would be better to include a sequence comparison between HCAR2 and HCAR3, as well as with P2Y1 in the region critical for activation, to make the conclusion more reliable.*

We added sequence alignments of HCAR2 with HCAR1, HCAR3, ADRB2 and P2Y1. Please refer to Supplementary Figure 7.

2-2. *Additionally, I'm not sure if P2Y1 is a good choice for the inactive structure, some other reported structures from the δ branch of Class A GPCR or the recently published crystal structure of HCAR2 might be considered.*

We have changed the figure in question, and now use the recently published crystal structure as suggested. Please refer to Figure 5b-e.

3. *In line 116 and Supplementary Fig.4, the authors mention that "the extracellular face shows a distinct pattern of electrostatic". It is unclear if this phenomenon is specific to HACRs and whether this feature has any special impact on its functionality.*

This is a general phenomenon, that allows proteins to bind particular ligands or groups of ligands, but in the case of HCARS it also controls the response of the receptor to ligand binding. Due to the difficulty of clearly depicting the electrostatic surface, we have modified figure 2d to show the molecular surface colored by hydrophobicity instead. Additionally, we have revised the text that confused the reviewer.

4. *The Results session of "Orthosteric binding site" is quite long and may need to be better organized. For example, the comparison of HCAR2 with HCAR1/HCAR3 and the comparison between the three different ligands could be separated.*

We have split the "Orthosteric binding site" section into two, "Orthosteric binding site" and "Putative determinants of ligand selectivity between HCAR2 and HCAR3". Checking our paper against the recent Yang et al paper, we find they used a similar number of words (about 460) to describe recognition of MK-6892 as we used (about 500) to describe the binding of three separate ligands. Following a

suggestion of Reviewer 2 (see below) we have moved some sentences to the Discussion and reworded them.

5. *There are mutagenesis experiments in this manuscript, but the expression levels of the mutated constructs are lacking. It would be beneficial to include data on the key mutations to demonstrate that the mutations do not significantly affect expression levels. Without this information, the conclusions may not be convincing.*

We now show the expression levels of wild-type and mutant HCAR2 in Supplementary Figure 6a.

6. *Line 166-167, "suggesting that this tyrosine and TM7...controlling the balance of ..." this conclusion is quite strong, but I am not fully convinced by how this conclusion is supported. Especially the authors made an analogy with the case in CCR1, which I find quite confusing.*

We have reworded the relevant sentences. The role of the tyrosine is suggested by earlier work on CCR1 (cited), so that the size of ligand binding to CCR1 may control signal bias. We simply note a similar effect with HCAR2. If the tyrosine (Y284) is replaced by alanine then only the largest ligand tested gives increased Gi activity and reduced β -arrestin recruitment.

7. *The Molecular dynamics stimulation experiments were used to study the ligand entry, which is an interesting direction but certainly needs more experimental support.*

MD simulations were carried out again using the improved structure. As well as the wildtype, the R22W mutant was subjected to MD. The results of these studies were consistent with the experimental mutation results. The MD work was carried out to see whether this completely independent computational method also suggested one particular route of entry over another. We believe the fact that our mutational study and the MD study both suggest the same route is a valuable result.

8. *The Discussion session is too simple, and seems to be written in a rush. There are some interesting directions that could be discussed, for example, the ligand entry path, the selectivity, and signaling bias. What have been found through this study, which can be further investigated.*

In the recent Yang paper we find no Discussion section at all, only a final paragraph of under 100 words that concludes by suggesting the work should assist future ligand design. We have extended the Discussion of our manuscript, referring to the lack of an ionic lock and the possibility that flexibility of the extracellular loops may have important biological implications. Both of these aspects of HCAR2 are yet to be investigated by appropriate mutational studies.

9. *Line 293, the last sentence in the Discussion mentioned the side effect of flushing, however, it is not clear how this side effect is caused, is it due to lack of selectivity between different HCARS, or b-arrestin signaling, or other pathways? Without understanding this, I'm not convinced that this study could provide insight into design of drugs with reduced side effect.*

The Yang paper noted that the correlation between β -arrestin signaling, and skin flushing requires further research, and that is exactly what we have attempted to provide here. We have revised the wording of the Discussion to avoid overinterpretation of our study, and now cite the Yang paper when discussing the somewhat contradictory results of studies regarding flushing, emphasizing the need for further investigation of the mechanism underlying flushing through HCAR2.

10. *In Supplementary Fig.1, the band close to HCAR2 is labeled with Gai1-G γ 2, is it a fusion expression? I didn't see this in the Methods.*

To clarify the construction of G-protein heterotrimer, we added additional details to the Methods section.

11. In Supplementary Fig.2, the text overlaps each other, maybe due to the conversion from word to PDF.

We have modified Supplementary Figure 3 to correct this.

12. The writing of this manuscript needs more work to polish. A lot of typos and grammatical problems. Just to give a few examples: Line 261, "in presence niacin and acipimox"; Line 265-266, "but R128...has not such contribution", this sentence is confusing.

We have revised a number of typos and grammatical errors.

Reviewer #2 (Remarks to the Author):

1. The manuscript can be strengthened by adding a more comprehensive Discussion section, which can be focused on comparing results of this study with previous ones on similar systems, i.e. homologous GPCR in terms of structures and previous molecular/cell biology experiments on HCARs, for example. Current version of the "Discussion" section just provides some brief general overview of the study results. Also, there are some discussion elements embedded in the Results section, e.g. on lines 158-167 and 180-184.

We have followed the suggestion of moving the indicated lines to the Discussion section, which also addresses the complaints of Reviewer 1.

2-1. The description of NanoBiT assays and their results are somewhat confusing. Sometimes in the manuscript their results are referred to as "signaling" and sometimes as "binding" and "dissociation." This along with what quantities are being measured need to be clarified.

We have aligned the terminology used to refer to the NanoBiT assay results as "Gi activity" and "β-arrestin1 recruitment."

2-2. It would be great to report some quantities such as K_d , K_i or equivalent for easier comparison than plots if this is amenable. I was also surprised that ligand binding affinities or other similar quantities were not even mentioned in the study because they represent very important quantities for comparing different ligands and/or protein mutants. Another important consideration is comparison of beta-arrestin and G protein assays, which the authors only glanced over in the manuscript. However, this is a very important subject alluding to biased signaling, which can be analyzed and discussed much more (see comment 1 above).

We added graphs for pEC_{50} and E_{max} . Please refer to Supplementary Figure 1c, 6 and 9g.

3. There is no amino acid sequence information in the supporting text for HCAR2 and other related GPCR discussed. Sequence alignment and comparison will be very useful.

We added sequence alignment of HCAR2 with HCAR1, HCAR3, ADRB2 and P2Y1. Please refer to Supplementary Figure 7.

4. The description of molecular dynamics (MD) simulations and their analyses are somewhat confusing. It is not clear what kind of simulations were performed and whether convergence was achieved. It seems that they were largely unbiased runs with flat-bottom z position restraints used to prevent ligand flipping over to the intracellular side. Does it mean that spontaneous ligand binding was probed? Did a ligand ever reach an orthosteric site during microsecond-long MD runs? It is not clear from Figure 4D, which does not provide many useful insights. If a main purpose was sampling of initial extracellular binding as shown in Fig. 4A&C, this needs to be clarified as well. Also, some quantitative analysis

using RMSD from the orthosteric binding poses or interaction energies using e.g. MM-PBSA or similar approach could be potentially used.

We revised the text and added explanations of the MD simulations and their results. To find the ligand entry point, we employed conventional MD simulations started by placing the ligand away from the protein in the cell exterior (Supplementary Figure 9e) and sampled spontaneous initial binding events. The frequency of residues interacting with the ligand is shown in Figure 4a. Three residues H9, R22 and R270, showed high contact frequency (Supplementary Figure 9a), in both 500 ns and 1000 ns MD simulations. No biasing techniques were used. The ligand sometimes moved across the periodic boundary of the z direction (i.e., the upper direction of the unit cell shown in Supplementary Figure 9e). To prevent artifacts due to the periodic boundary condition, we used the flat-bottom z position restraints which added force only when the ligand was over 40 Å from the protein. As the reviewer commented, due to the timescale limitation, the ligand never reached the orthosteric site, even during 1 microsecond MD runs. This is clear from RMSD from the ligand bound within the orthosteric binding site, as well as the minimum distance between R111 and ligand (Supplementary Figures 9b and 9c). The snapshot with the lowest minimum distance in the MD runs was also showed in Supplementary Figure 9f. To show the effects of the mutation on the minimum distances clearly, we added the histogram of the minimum distances in the revised Figure 4d, showing that the minimum distances for the mutant were longer than those for wild type. This behavior was in good agreement with the experimental functional assays. These points are explained in the revised manuscript and figures.

5. *Molecular details of HCAR2 - Gi interactions as shown in Fig. 5F and G remain very unclear. It would be best to show specific inter-residue interactions e.g. in Fig. 5G, at least the most important ones. So, Gi alpha5 residues can be also schematically shown and their H bonding or other directional interactions with the receptor residues indicated e.g. by thin dashed lines.*

We have modified Figure. 5f and g for better clarity.

6. *Molecular docking of MK-1903 shown in Supplemental Fig. 6 is not very clear in terms of purpose, methodology etc. Docking methods should be briefly mentioned. More importantly, it is not clear what is being compared. If it is selectivity against HCAR3 as indicated on lines 189-190 of main text, then docking to homology or de novo (e.g., AlphaFold) model of HCAR3 would have also needed to be done. Otherwise, it is hard to conclude that "it seems highly likely that selectivity is achieved through the replacement of S178ECL2 with isoleucine. Isoleucine in this position cannot adopt a rotamer that does not strongly clash with MK-1903" Also, a comparison of AlphaFold model of HCAR2, used as a template for density fitting, with solved cryo-EM structures would be interesting to show as well.*

We carried out the *in-silico* MK-1903 docking using the widely used software Glide and the improved structure of HCAR2. The detailed methodology has been added to the Method section. We discussed the selectivity based on the pose with best score. We have moderated the language and now suggest only that some selectivity is achieved through the serine to leucine mutation. Additionally, we aimed to identify possible factors for ligand selectivity by examining a model of HCAR3 generated by multi-state GPCR modeling using AlphaFold2 and the structure of HCAR2.

7. *A lot of references are not listed or missing, especially for the Methods section. Please check.*

We have added references in the Methods section.

8. *There are a number of minor issues as detailed below as well (with some overlap / duplications with ones above possible):*

We have revised a number of typos and grammatical problems.

Reviewer #3 (Remarks to the Author):

The manuscript, “Structural basis for ligand recognition and signaling of hydroxy-carboxylic acid receptor 2” by Jae-Hyun Park et al. presents 3 cryo-EM structures of HCAR2 bound to different small molecule ligands. The cryo-EM structures are supported by mutagenesis and pharmacology data that provide insight into how the three different ligands bind to the orthosteric site of HCAR2, which is buried deep within the receptor. Molecular dynamics simulations and mutant/pharmacology data provide insight into a unique mechanism of how these ligands access the receptor. A comparison of the activation mechanism of HCAR2 and its interactions with G proteins were performed by comparison to the inactive P2Y1 receptor structure (closest homology structure to HCAR2).

Overall, the manuscript is well written and presented. A minor concern is that there isn't much mechanistic insight provided by this manuscript, other than how these three ligands bind HCAR2. The insight into how agonists access the orthosteric site and the activation mechanisms are rather speculative at this point and not supported by convincing experimental data.

The much larger and more significant concern is the quality of the cryo-EM models and cryo-EM maps. The provided PDB validation reports provide evidence that the modelling of the receptor is rather poor by comparison to what would be expected. Presumably, the authors would have checked over this report in detail and realized the structures needed improvement prior to submission. There are a relatively large number of significant clashes, poorly modelled protein backbones, and protein sidechains all on the receptor chain. The Q-score indicates that there could be issues with resolvability of atoms around the receptor region (and parts of the G protein).

*As suggested by the reviewer we have reprocessed the cryo-EM data sets, resulting in improved electron density maps, models and the Q-score compared to the previous ones. We believe that there have been significant improvements, especially in the region of N-terminus, TM1 and TM2. The final PDB, map files and validation reports are available to download using the following URL:
<https://www.dropbox.com/scl/fo/fj6fedidhyqylzhuhmgf2/h?rlkey=flan9ekqslzq65dyidv5ydmhf&dl=0>*

These concerns are heightened when looking at the provided experimental maps and models. For example, residue TYR284 is suggested to be in a different rotamer conformation in the GSK256073 structure (Fig 5B), however, it's unclear how the authors can make this claim, as evidenced in the provided screenshots in the pdf version of this review. The residue could in fact be in the same conformation as the other structures as there is convincing EM-density for it. The authors have instead modelled part of GSK here, which doesn't really seem plausible as the aliphatic chain would not have density in this shape and is instead likely to be unstructured/dynamic manner. Nevertheless, the quality of the map/models prevents a clear conclusion here.

Upon detailed examination during building the structure, we discovered that, as pointed out by the reviewer, the GSK256073 was incorrectly built in. In the revised structure, we have built it in the appropriate pose. Based on the newly modelled structure, we have revised all relevant content.

Similarly, residue W188 in the GSK-bound structure looks like it is modelled into the noise of the map, rather than a different conformation as shown in Figure 2E. These are just a few examples. The extracellular regions are also rather messy and it is unclear how reliable the modelling would be in these regions. The authors have modelled large portions of the Galpha subunit into the noise of the map and in some cases there is no density present to support any protein modelling at all. These are all significant issues that need to be addressed, and the best place to start would be improving the quality of the cryo-EM maps. The sharpened maps that were provided appear to be over sharpened and the deep EM enhance maps are not very reliable around the regions of the receptor, which is the key region of the manuscript. I'd suggest the authors perform a local refinement around the receptor only, which should be easy to do given the data was processed in CryoSparc. Hopefully, these maps will be of better

quality than the provided maps, which would significantly improve the modelling of the receptor. Otherwise, the authors made need to adjust the data processing strategies to try and improve quality of the maps (e.g. they could process the data in Relion, where 3D classifications and polishing can make a considerable improvements in the maps versus cryosparc). Once the maps are improved, the authors will need to spend time improving the modelling on all regions of the GPCR-G protein complex, and then spend time reinterpreting their main findings.

Following the reviewer's comment, we performed local refinements, resulting in a significant improvement in the electron density maps. We have used the newly modeled structures to interpret and write the revised content accordingly. We have removed the portions of the model corresponding to the HCAR2 and Galpha subunit where the electron density was not visible.

Other comments:

1) *The authors should include a figure that shows the local resolution of their cryo-maps.*

We have added a figure showing the local resolution in Supplementary Figure. 3

2) *It's not made clear in the text why the authors performed the B-arrestin assays? I was expecting a discussion about bias or something, but in the end was just left wondering why the data was even there.*

We believe the amended text, especially the Discussion, makes clear the need for this type of data to unravel the relationship between HCAR2 and the flushing response.

3) *Figure 4 and the MD simulations. The quality of the structures does not support any of the data here, residues R22 and K16 are modelled into nothing/noise. As mentioned previously, the quality of the cryo-EM density around the extracellular region is rather poor, as is the modelling. I'm not sure how one can draw any conclusions from this region of the model, and the complementary MD data isn't very convincing. For example, in the comparison in Fig 4D with WT and R22W, the minimum distance is achieved in 1-2 simulations in a very transient manner. This whole section should be removed.*

Many residues, including R22 and K16, are now appropriately modeled based on the improved electron density map. The mentioned residues, R22 and K16, have clear electron density now. All of the MD simulations were carried out again from the improved structure. The ligand entry pathway of the initial interacting residues was analyzed again. The results of the new MD simulations were not essentially different from the previous one. Unlike the ligand binding pose at the orthosteric site, the ligand binding poses in the encounter structures were not stable because they did not form strong anchoring interactions. The binding and dissociation were repeated, but the pattern of interacting residues on the extracellular side did not change between 500 ns and 1000 ns, indicating convergence of the contact frequencies (Supplementary Figure 8a). In addition, the distribution of the minimum distances between the ligand and protein is clearly different between the wildtype and the R22W mutant (Figure 4d), and the fact that the ligand was less likely to enter the mutant protein than the wildtype is consistent with the experimental results. Because the new MD simulations using the refined structures match the experimental results, this section was not removed in the revised manuscript.

4) *With regards to the activation mechanism, this of course difficult to do in the absence of an inactive state structure. I'm sure the editor's and authors are aware of the HCAR2 structures that were recently published in Nature Communications, which include inactive state structures. The authors should be comparing their work to those structures (both inactive and active). For example, are there any differences between the studies, other than the orthosteric ligands?*

As comment by the reviewer, we have changed the figure that compares our structures with the recently published HCAR2 structures.

5) *Does the inclusion of BRIL to N-terminus affect the pharmacology of the receptor? Given the position of the N-terminus over the top of the receptor it would be prudent to perform these experiments.*

The N-terminus BRIL conjugated HCAR2 construct showed intact Gi and β arr1 response to the three ligands. Please refer to revised Supplementary Figure. 1b and 1c

6) *More details for Supp Fig 3 need to be provided. What maps are shown, what contour level, and was the density carved or masked and if so what was the resolution cut-off (i.e. provide enough that this figure can be recreated by others).*

In the figure legend, we added the details used to create Supplementary Figure 4.

Reviewers' Comments:

Reviewer #1:

Remarks to the Author:

The authors addressed my major concerns and I agree with the publication.

Reviewer #2:

Remarks to the Author:

The manuscript describes new cryo-EM structures of hydroxycarboxylic acid receptor 2 (HCAR2), a class A G protein-coupled receptor (GPCR) bound to 3 agonists (niacin, acipimox or GSK256073) as well as inhibitory G protein (G_i). The authors not only solved those structures, but also performed NanoBiT G-protein dissociation and beta-arrestin assays on wild-type and multiple mutant HCAR2s to probe determined ligand binding pockets as well as putative binding pathways from all-atom molecular dynamics simulations. The authors found some unique structural features of HCAR2, its ligand binding and G protein activation, which may help develop new drug molecules for human disorders such as dyslipidemia and inflammation.

This is the revised manuscript and the authors did a great job addressing reviewers' suggestions. I think the manuscript is almost ready for publication aside from some minor issues indicated below.

In the revised manuscript on lines 123-124 the authors said that "electron density within this pocket suggests the presence of cholesterol"

How was it determined that it is cholesterol and is its binding important for the receptor activation and/or drug binding? This is never elaborated in the manuscript.

The authors performed 2 NanoBiT assays, one for G protein dissociation and the 2nd for beta-arrestin recruitment. Yet, their cryo-EM study focused exclusively on G_i but not beta-arrestin signaling. This is briefly discussed but may need to be elaborated more in the Discussion section. Of note, there is a paper, which just came out in Molecular Cell journal (<https://doi.org/10.1016/j.molcel.2023.07.030>), which addresses this issue more explicitly. Since it came after the revised version of the manuscript was received, this should not be a problem, but a brief mentioning of biased agonism and allosteric modulation may be helpful to indicate future directions. Also, I am very glad that the author added figures showing EC50 and Emax plots, but it would be also great if numerical values in the tabular form would be provided as well.

Also, for MD simulations, it makes sense that unbiased MD simulations were too short to see ligand binding to such enclosed pockets, but the authors might also mention alternative ways to explore in future studies such as well-tempered metadynamics (e.g., used in a recent study for beta-adrenergic receptor - ligand binding pathways, Xu et al Cell Res 2021 May;31(5):569-579. doi: 10.1038/s41422-020-00424-2), Gaussian accelerated MD or other enhanced sampling techniques.

There is also a list of more specific small comments/suggestions below:

I 103 "G_i1 and βarrestin1" -> please use consistent notation, G_i or G_i1, β-arrestin or βarrestin1. And please spell out the 1st time those and other terms are used. Also, for "G_i" throughout the manuscript, it is typically written as G_i (subscript),

I 111 "TM1" - please spell out

I 127 rmsd - please capitalize and spell out

I 201 "One of these is formed by K15Nterm, K16Nterm, K1654.63 and K1664.64, and the other is near H9Nterm/R221.27 (Figure 2d)." - only R22 is shown in Fig. 2d. Please double check.

I 228 "P2Y1" - please spell out

I 256 "activation structures" -> active or activated structures"

II 320-327 Please indicate what black vs colored curves mean in Fig 1a plots

I 322 "SEM" - please spell out

I 328 "Structure features" -> "Structural features"

I 330-331 "Three disulphide bonds and proline residues, which supply rigidity on the extracellular face of HCAR2, are shown as pink cartoon and stick models" - Some loops are also shown as pink, not only prolines. Please clarify

I 333 "ELC1 and ECL2" -> "ECL1 and ECL2"

I 334 "hydrophobic and hydrophilic areas are shown as yellow and red" -> "hydrophobic and hydrophilic areas are shown as yellow and orange"

II 342-345 Please indicate what type of interactions are shown by LigPlot+ using semi-circles.

I 357 Please indicate what kind of distances you measured, between particular atoms, centers of mass, geometry etc.

I 362 "the on niacin bound HCAR2 structure" -> "on the niacin bound HCAR2 structure"

I 376 "Atoms used for angle (black line) calculation are indicated in black." Please specify which specific angle you mean

I 414-415 "of PH promoter of pFastbac dual vector" - please explain what is PH promoter

I 588-591 "Then, in the niacin and GSK256073 systems, the HCAR2 structure was replaced to that of the Acipimox system... Initial models of HCAR2 with niacin was [were] prepared from the complex solved by cryo-EM.". There is a contradiction here, please correct

II 596-597 "D73 and E37 were set to a protonated aspartic and glutamic 597 acid, respectively. The N ϵ atom in H134 was protonated" - please indicate citation or something else on why D73 and E73 or their equivalents in other GPCRs were protonated. Also, please indicate protonation states of all His residues. For H134 it is vague as N ϵ can be protonated in neutral HSE and cationic HSP (with N δ protonated as well).

I 599 "Supplementary Fig. 7c" - this does not exist

I 608 "potassium and chloride counterions" since we are looking into ligand binding on the extracellular site, it would be more appropriate to use 0,15 M NaCl solution, although likely there is no big difference

I 632 "electrostatic interaction was" -> "electrostatic interactions were"

I 634 "The van der Waals interaction was" -> "The van der Waals interactions were""

I 648 "a grid of cube" -> "a cubic grid"

I 651 "grid potential parameter was calculated without the ligand" - this is confusing, please clarify

I 652 "obtained by" -> "obtained from"

I 656 "shown in Supplementary Fig. 7." -> "shown in Supplementary Fig. 8."

I 725 "chimeraX" -> "ChimeraX"

I 729 Please spell out names of GPCRs when possible

I 731 "red" -> "orange"

II 737-744 Please indicate what EC50 and Emax mean. They were defined in the main text but not in SI.

I 749 "BW numbers" please spell out

I 753 "Molecular docking model of MK-1903 on the HCAR2 structure"
Please specify which HCAR2 structure was used.

II 768-769 Panel g results are not part of MD simulations. Please put it as a separate figure or amend the title e.g. as "MD simulations of ligand binding pathways and supporting experimental mutagenesis results"

I 778 Please spell out "MT1" and indicate PDB ID

Fig. 1b I would indicate with arrows ligand binding sites in the structures, otherwise their separate densities are somewhat confusing.

Fig. 2. It would be good to indicate cholesterol and ligand binding sites with respect to the overall receptor structure, e.g. by indicating them on panels a or d. Also, labeling TM1-7 on panel A would be helpful to orient the reader.

Fig 4d. "Number of snapshot" is somewhat counterintuitive. I would change this axis to probability by dividing it to the total number of MD frames used for the analysis. Also, in my opinion, lower values, i.e. tails in these distributions matter much more and might need to be shown as an inset.

Reviewer #3:

Remarks to the Author:

The authors have adequately addressed my main concern of the manuscript, which was the quality of the cryo-EM maps and model. The consensus map is still a bit difficult to interpret, but the local refinement maps are improved to a sufficient level to support the main claims of the manuscript. My other concerns were appropriately addressed as well, and the overall revisions have improved the quality of the manuscript.

Two minor comments:

- 1) Lines 110-11 should be revised to include the N-terminal regions of G β and G γ as regions without clear electron density.
- 2) Electron density is not the correct term for density from a cryo-EM structure, an alternative suggestion would be cryo-EM density.

Response to reviewers

We thank the reviewers for their valuable comments, which have significantly contributed to improving the manuscript. Please find the final revised version, which has been thoroughly revised in accordance with the referee comments. Changes to the text have been highlighted in red. We have also provided point by point responses to each referee's comments and suggestions.

Reviewer #2 (Remarks to the Author):

1. *In the revised manuscript on lines 123-124 the authors said that "electron density within this pocket suggests the presence of cholesterol"*

How was it determined that it is cholesterol and is its binding important for the receptor activation and/or drug binding? This is never elaborated in the manuscript.

The residues of HCAR2 that form this pocket are highly similar to cholesterol binding site 2 of CX3CR1, as mentioned in the manuscript (citing ref. 21). Additionally, among the compounds included in the sample used for structure determination, only cholesterol has the appropriate chemical properties and sufficient size to account for this cryo-EM density. These are sufficient grounds to speculate that this cryo-EM density represents a putative cholesterol binding site. In the paper just published in Molecular Cell journal (mentioned below; <https://doi.org/10.1016/j.molcel.2023.07.030>), the authors place cholesterol in same pocket proposed in this manuscript. We have revised the corresponding sentence to make it clearer.

We did not assess the effect of these residues on HCAR2 activation; however, in the case of CX3CR1, some residues forming this site contribute to receptor activation. Therefore, these residues of HCAR2 may also play a role in receptor activation or signaling.

If cholesterol binding at this site does indeed influence HCAR2's activation, it may be possible to design allosteric modulators targeting this site, although HCAR2-specificity might not be guaranteed. (Allosteric modulators such as AP8 and compound 6FA target region near cholesterol binding site, within TM3, TM4 and TM5 in GPR40 and beta2AR, respectively).

2. *The authors performed 2 NanoBiT assays, one for G protein dissociation and the 2nd for beta-arrestin recruitment. Yet, their cryo-EM study focused exclusively on Gi but not beta-arrestin signaling. This is briefly discussed but may need to be elaborated more in the Discussion section. Of note, there is a paper, which just came out in Molecular Cell journal (<https://doi.org/10.1016/j.molcel.2023.07.030>), which addresses this issue more explicitly. Since it came after the revised version of the manuscript was received, this should not be a problem, but a brief mentioning of biased agonism and allosteric modulation may be helpful to indicate future directions.*

We have added a few additional sentences about biased agonism and allosteric modulation in

Discussion section.

3. Also, I am very glad that the author added figures showing EC50 and Emax plots, but it would be also great if numerical values in the tabular form would be provided as well.

We have added a table for pEC₅₀ and E_{max}. Please see Supplementary Table 3.

4. Also, for MD simulations, it makes sense that unbiased MD simulations were too short to see ligand binding to such enclosed pockets, but the authors might also mention alternative ways to explore in future studies such as well-tempered metadynamics (e.g., used in a recent study for beta-adrenergic receptor - ligand binding pathways, Xu et al *Cell Res* 2021 May;31(5):569-579. doi: 10.1038/s41422-020-00424-2), Gaussian accelerated MD or other enhanced sampling techniques.

Following the reviewer's comment, we added a discussion about the timescale limitation in our simulations and future plan for sampling the entire ligand-binding pathway to the end of the HCAR2 ligand binding pathway section, as follows,

"As discussed above, the ligand entry pathway up to the encounter structure at the surface of the protein (Figure 4d), which corresponds to the initial stage of the ligand binding pathway, could be verified in the conventional MD and the mutant experiments. However, there was a still limitation in the timescale of conventional MD to reveal the entire binding pathway from the surface to the internal orthosteric site of the protein. The introduction of enhanced sampling techniques with MD, such as well-tempered metadynamics which has been recently applied to the ligand binding pathway for GPCRs^{23,24} or accelerated MDs^{25, 26} might help to reveal the entire ligand binding pathway, and thereby afford better understanding of the dynamics of activation."

23. Saleh, N. et al. An Efficient Metadynamics-Based Protocol To Model the Binding Affinity and the Transition State Ensemble of G-Protein-Coupled Receptor Ligands. *J. Chem. Inf. Model* 57, 1210-1217 (2017).

24. Xu, X. et al. Binding pathway determines norepinephrine selectivity for the human β 1AR over β 2AR *Cell Research* 31, 569-579 (2021).

25. Hamelberg, D. et al. Accelerated molecular dynamics: A promising and efficient simulation method for biomolecules. *J. Chem. Phys.* 120, 11919-11929 (2004).

26. Miao, Y. et al. Gaussian Accelerated Molecular Dynamics: Unconstrained Enhanced Sampling and Free Energy Calculation. *J. Chem. Theory Comput.* 11, 3584-3595. (2015).

5. There is also a list of more specific small comments/suggestions below:

l 357 Please indicate what kind of distances you measured, between particular atoms, centers of mass, geometry etc.

We revised the explanation as

"Distributions of the minimum distance between the nearest non-hydrogen atoms of niacin

and R111^{3,36}”.

588-591 *"Then, in the niacin and GSK256073 systems, the HCAR2 structure was replaced to that of the Acipimox system... Initial models of HCAR2 with niacin was [were] prepared from the complex solved by cryo-EM."*. There is a contradiction here, please correct.

The sentence “Initial models of ... “ is superfluous, and the immediately preceding sentence correctly describes the initial structure setup, so the superfluous sentence has been removed.

l1 596-597 *"D73 and E37 were set to a protonated aspartic and glutamic acid, respectively. The N ϵ atom in H134 was protonated" - please indicate citation or something else on why D73 and E73 or their equivalents in other GPCRs were protonated. Also, please indicate protonation states of all His residues. For H134 it is vague as N ϵ can be protonated in neutral HSE and cationic HSP (with N δ protonated as well).*

The section describing the protonation state has been revised to clearly state that the protonation state was set according to PROPKA’s predictions, and also to show the protonation state of all histidine residues, as follows:

“The protonation states of the protein, charged residues and histidine residues at pH 7 were set according to the PROPKA estimation implemented in PDB2PQR⁵⁴: D73 and E37 were set to a protonated aspartic and glutamic acid, respectively. H134 was set to the neutral histidine with only the protonated N ϵ atom (HSE in the CHARMM force field), while other histidine residues were set to the neutral histidine with only the protonated N δ atom (HSD in the CHARMM force field).”.

l 599 *"Supplementary Fig. 7c" - this does not exist*

We removed Supplementary Fig. 7c, because we revised Supplementary Figure 8 to include all the information of Supplementary Fig. 7c of the previous revision.

l 608 *"potassium and chloride counterions" since we are looking into ligand binding on the extracellular site, it would be more appropriate to use 0,15 M NaCl solution, although likely there is no big difference*

Thank you for pointing out. We agree that the appropriate biochemically relevant ion is 150 mM NaCl. As for the MD results, we believe the results with 150 mM KCl are considered invariant to be those with 150 mM NaCl because no ions appear to be directly involved in the protein dynamics.

Fig 4d. "Number of snapshot" is somewhat counterintuitive. I would change this axis to probability by dividing it to the total number of MD frames used for the analysis. Also, in my opinion, lower values, i.e. tails in these distributions matter much more and might need to be shown as an inset.

We changed the vertical axis in Fig. 4d to represent probability. In addition, following the reviewer's suggestion, we analyzed the tail region at the lower value to understand how far the ligand reached the binding site. The shortest distances for wild-type and the R22^{1,27}W mutant are 16.2 Å and 15.9 Å, respectively, and the probability for the mutant was slightly higher than that for the wild-type. However, because the probability (under 0.001 corresponding to 3 snapshots in the total 3000 snapshots) was very low, it was considered that a statistical discussion in the tail region was not feasible, and thus, we did not add the inset illustrating the tail to the Figure 4d.

1 753 "*Molecular docking model of MK-1903 on the HCAR2 structure*"
Please specify which HCAR2 structure was used.

We revised the part as “the acipimox bound HCAR2”.

All other small comments were corrected in the main text and, figures.

Reviewer #3 (Remarks to the Author):

1. Lines 110-11 should be revised to include the N-terminal regions of G β and G γ as regions without clear electron density.

We have changed the main text.

2. Electron density is not the correct term for density from a cryo-EM structure, an alternative suggestion would be cryo-EM density.

Thank you very much for your comment.